# An Alba-domain protein required for proteome remodelling during trypanosome differentiation and host transition

**Shubha Bevkal**[1,2], **Arunasalam Naguleswaran**[1], **Ruth Rehmann**[1], **Marcel Kaiser**[3,4], **Manfred Heller**[5], **Isabel Roditi**[1]*

**1** Institute of Cell Biology, University of Bern, Bern, Switzerland, **2** Graduate School of Cellular and Biomedical Science, University of Bern, Bern, Switzerland, **3** Department of Medical and Parasitology and Infection Biology, Swiss Tropical and Public Health Institute, Basel, Switzerland, **4** University of Basel, Basel, Switzerland, **5** Proteomics and Mass Spectrometry Core Facility, Department for BioMedical Research, University of Bern, Bern, Switzerland

* isabel.roditi@izb.unibe.ch

**Data Availability Statement:** Raw read files for RNA-seq data are deposited at the European Nucleotide Archives (ENA) under study

## Abstract

The transition between hosts is a challenge for digenetic parasites as it is unpredictable. For *Trypanosoma brucei* subspecies, which are disseminated by tsetse flies, adaptation to the new host requires differentiation of stumpy forms picked up from mammals to procyclic forms in the fly midgut. Here we show that the Alba-domain protein Alba3 is not essential for mammalian slender forms, nor is it required for differentiation of slender to stumpy forms in culture or in mice. It is crucial, however, for the development of *T. brucei* procyclic forms during the host transition. While steady state levels of mRNAs in differentiating cells are barely affected by the loss of Alba3, there are major repercussions for the proteome. Mechanistically, Alba3 aids differentiation by rapidly releasing stumpy forms from translational repression and stimulating polysome formation. In its absence, parasites fail to remodel their proteome appropriately, lack components of the mitochondrial respiratory chain and show reduced infection of tsetse. Interestingly, Alba3 and the closely related Alba4 are functionally redundant in slender forms, but Alba4 cannot compensate for the lack of Alba3 during differentiation from the stumpy to the procyclic form. We postulate that Alba-domain proteins play similar roles in regulating translation in other protozoan parasites, in particular during life-cycle and host transitions.

## Author summary

*Trypanosoma brucei* is a unicellular eukaryotic parasite that is responsible for African trypanosomiasis. The parasite needs two hosts, mammals and tsetse flies, in order to complete its life cycle. Throughout its developmental cycle, *T. brucei* encounters diverse environments to which it has to adapt in order to maintain its transmission and infectivity. Successful adaptation to the new environment and transition to different life-cycle stages are the general challenges faced by many digenetic parasites. In this study we show that the Alba-domain protein Alba3 is essential for differentiation of the mammalian

PRJEB38690. All other relevant data are within the manuscript and its Supporting Information files.

**Funding:** This research was funded by the Swiss National Science Foundation (www.snf.ch); Grant nos. 31003A_166427 and 310030_184669 to I.R. The funders had no role in study design, data collection and analysis, decision to publish, or preparation of the manuscript.

**Competing interests:** The authors have declared that no competing interests exist.

stumpy form (transition form) to the procyclic form in the tsetse host. An Alba3 deletion mutant infects mice and shows characteristic waves of parasitaemia, but is severely compromised in its ability to infect tsetse flies. Stumpy forms are translationally repressed, but are poised to resume protein synthesis during differentiation. We show that Alba3 is key to efficient escape from translation repression; in its absence, there is a delay in the formation of polysomes and resumption of protein synthesis. This impacts the formation of procyclic-specific mitochondrial respiratory complex proteins as well as the repression of some bloodstream-specific proteins. This is the first time that a single protein has been shown to have a major influence on translation as an adaptive response to changing hosts. It is also the first time that a mechanism has been established for Alba-domain proteins in parasites.

## Introduction

Digenetic parasites that cycle between mammals and arthropods encounter the most radical changes in environment during their movement between hosts. The generation of preadapted transmission forms is a critical step in ensuring efficient parasite dissemination to the other host. Since it cannot be predicted when transmission will occur, it is also vital that a parasite can respond rapidly to its new surroundings. This often entails differentiation to another developmental stage which differs markedly from the transmission form in its morphology, surface architecture and metabolism.

*Trypanosoma brucei* ssp. which cause human and animal trypanosomiasis, require both mammals and tsetse flies to complete their life cycle. The tsetse fly is the definitive host while mammals are intermediate hosts [1–4]. The two life-cycle stages found in mammals—proliferating slender forms and cell-cycle arrested stumpy forms, are covered by a variant surface glycoprotein (VSG) coat [5]. Trypanosomes escape the host immune response by periodically replacing the expressed VSG by an antigenically distinct one [6]. This, together with a quorum sensing mechanism which drives the differentiation of slender to stumpy forms [7–9] gives rise to the characteristic waves of parasitaemia in the bloodstream.

The generation of stumpy forms is crucial as it limits the parasite titre in the bloodstream, promoting host survival and sustaining the infection [10]. Stumpy forms are also preadapted for uptake by tsetse, with a more elaborate mitochondrion than slender forms [11,12]. With a life-span of only a few days in the mammal [13,14], however, their sole chance of survival is by onward transmission and further development in the fly. A tyrosine phosphatase, TbPTP1, and its substrate TbPIP39, as well as the kinases RDK1 and RDK2, have been identified as regulators of differentiation [15–17]. Knockdown of these factors or chemical inhibition causes bloodstream forms to express procyclic form markers in the absence of the normal *in vitro* differentiation triggers.

Upon ingestion by a tsetse fly, or exposure to differentiation signals in culture, stumpy forms develop into early procyclic forms. During this process they shed the VSG coat and replace it by EP and GPEET procyclins [18–20]. This is accompanied by enlargement and activation of the mitochondrion [11,21–23] and expression of mitochondrial proteins such as cytochrome oxidase complex proteins and ATP synthase subunits [24–27]. Over several days, early procyclic forms differentiate to late (GPEET-negative) procyclic forms. Transcriptomic and proteomic analyses suggest that the two forms have different metabolic capabilities: early procyclic forms express high affinity glucose transporters and glycolytic enzymes, while late procyclic forms upregulate enzymes involved in proline uptake and catabolism [28,29]. The

remainder of the cycle comprises a less well-characterised series of migration and differentiation steps in which the trypanosomes first move to the proventriculus, then colonise the salivary glands finally differentiating into infectious metacyclic forms that are transferred to a mammal when the tsetse takes a blood meal.

Most protein-coding genes in *T. brucei* are organized as polycistronic transcription units. The VSG and procyclin transcription units are transcribed by RNA polymerase I, and transcription is stage-regulated. All other protein-coding genes are transcribed by RNA polymerase II. There is no evidence for regulation of transcription initiation [30], so regulation of individual genes takes place at the level of mRNA processing, export, mRNA stability, translation and protein stability [31]. Consistent with the trypanosome's reliance on post-transcriptional mechanisms, RNA binding proteins have been found to play important roles in trypanosome developmental biology [31–33].

Alba-domain proteins were initially discovered as nucleic acid binding proteins in hyperthermophilic archaea, but they are also widely distributed in eukaryotes [34]. These proteins exhibit unusual functional plasticity. In archaea they play a role in chromatin organisation [35,36], while in yeast and mammals they are proposed to be functional equivalents of the RNaseP/MRP subunits Rpp20/Pop7 and Rpp25/Pop6 that are involved in tRNA and rRNA processing [37]. *T. brucei* encodes four Alba-domain proteins [38]. All four interact with components of the translation machinery and can be recruited to cytoplasmic starvation granules, where they colocalise with poly (A+) RNA. Immunoprecipitation studies revealed that Alba3 interacts with both Alba1 and Alba2. Knockdown of Alba3 caused slow growth of procyclic forms and also resulted in co-depletion of Alba1 and Alba2 proteins [38]. Alba3 and Alba4 are expressed throughout development in the tsetse fly, except during the mesocyclic to epimastigote transition in the proventriculus [39]. Interestingly, overexpression of Alba3 perturbs this transition [39].

Alba-domain proteins have been studied in other protozoan parasites. In *Leishmania infantum*, two Alba proteins shuttle between the cytoplasm, nucleus and flagellum during promastigote to amastigote differentiation. Deletion of LiAlba1 in amastigotes resulted in downregulation of developmentally regulated transcripts [40]. Knockout of TgAlba1 and TgAlba2 in *Toxoplasma gondii* severely impaired the parasite's ability to respond to stress and to differentiate [41]. It was also shown that translation of the TgAlba2 transcript required both its 3' UTR and the presence of the TgAlba1 protein. In *Plasmodium berghei*, the development of the zygote to the ookinete depends on the translational activation of stored, silent mRNAs in the P granules of female gametocytes, where PbAlba proteins are also found [42,43]. In addition, Alba proteins in *P. falciparum* interact with specific DNA sequences that are present at chromosome ends and play a role in the regulation of virulence genes [44]. Another study suggested a function for PfAlba1 in post-transcriptional gene regulation in blood stages [45]. In *P. yoelii*, PyAlba4 regulates gametocytes and sporozoites through stage-specific interactions and specific mRNA fates [46]. For the most part, however, nothing is known about the mechanism(s) by which the Alba-domain proteins exert their effects.

Here we describe deletion mutants of two Alba-domain proteins from *T. brucei*, Alba3 and Alba4, and the requirements for them in bloodstream forms and during the transition to procyclic forms. We show that Alba3 and Alba4 are functionally redundant in bloodstream forms. By contrast, only Alba3 can support differentiation from the stumpy to the procyclic form. This is achieved by extensive remodelling of the proteome.

## Results

### Alba proteins in *T. brucei*

The genome of *T. brucei* encodes four Alba-domain proteins: Alba1 (Tb927.11.4460), Alba2 (Tb927.11.4450), Alba3 (Tb927.4.2040) and Alba4 (Tb927.4.2030). In addition to the ALBA domain, Alba3 and Alba4 also contain nucleic acid-binding RGG/RG motifs in their C-termini (S1A Fig). Among them, Alba3 and Alba4 are most closely related to each other with 79.6% amino acid identity (S1B Fig). Previous work showed that all four Albas are cytoplasmic in early procyclic forms [38]. To ascertain if the localisation varies in the four life-cycle stages amenable to culture (slender and stumpy bloodstream forms, early and late procyclic forms), individual Alba proteins were tagged in situ with an N-terminal haemagglutinin (HA) tag.

Immunofluorescence analysis confirmed that all Albas were predominantly cytoplasmic in all four stages (S2 Fig). In stumpy forms, however, Alba3 and Alba4 showed an additional spot close to the kinetoplast (Fig 1A). Recently, it was shown that regulators of differentiation, including TbPIP39, accumulated at a location that was named the "stumpy regulatory nexus" (STuRN), [47]. We performed co-immunostaining for Alba3/Alba4 and TbPIP39. Fig 1B shows that they colocalise, indicating that Alba3 and Alba4 are also present in the STuRN.

In previous studies, depletion of Alba proteins by RNAi was performed only in procyclic forms [38,39]. To evaluate their importance in other life-cycle stages, we performed RNAi in slender bloodstream forms and early and late procyclic forms of the pleomorphic line AnTat90-13. For these experiments we used the RNAi constructs designed by Mani and coworkers to knock down Alba transcripts [38]. Individual knockdown of Alba1, 2 or 4 did not cause a growth defect in any life-cycle stage, nor did simultaneous knockdown of Alba1+2 (S3 Fig). However, knockdown of Alba3 resulted in slower growth in all three life-cycle stages (Fig 1C). Simultaneous knockdown of Alba3+4 gave similar results to Alba3 alone in slender forms and early procyclic forms, but intensified the defect in late procyclic forms (Fig 1D). Interestingly, and in contrast to what was observed with early procyclic forms ([38] and Fig 1D), Alba4 consistently escaped knockdown in Alba3+4 double RNAi in slender bloodstream forms. In summary, these data suggest that Alba3 plays a key role in several life-cycle stages. Therefore, we decided to characterise Alba3 in detail.

### Alba3 is not essential in bloodstream forms, but is required by procyclic forms

Since knockdown by RNAi is never complete, we examined the essentiality of Alba3 by a classical knockout (KO) approach (S4A Fig). This was performed in slender bloodstream AnTat 90–13. PCR and western blot analysis of Alba3KO confirmed that the gene was deleted (S4B Fig) and the protein was not detectable (S4C Fig). Alba3KO bloodstream forms showed slightly slower growth, with a population doubling time of 6.42h (±0.12h) compared to the parental line (6h, ±0.09h), but this difference was not statistically significant (p = 0.09; Fig 2A). As assessed by morphology and expression of the stumpy-specific marker PAD1, KO slender forms retained the ability to differentiate to stumpy forms at rates comparable to the parental line (Fig 2B and 2C). We next compared the ability of stumpy forms to differentiate to procyclic forms. When exposed to cis-aconitate and a drop in temperature to 27˚C, the parental line re-entered the cell cycle after 15h (Fig 2D) and grew exponentially after 24h (Fig 2E). Alba3KO re-entered the cell cycle at 18h (Fig 2D) and also grew exponentially, but more slowly for the first few days. By day 5, however, Alba3KO stopped dividing (Fig 2E). To ensure that this defect was due to deletion of Alba3, an addback was constructed by stably transforming the KO with an inducible expression construct (Alba3cKO). In the absence of tetracycline,

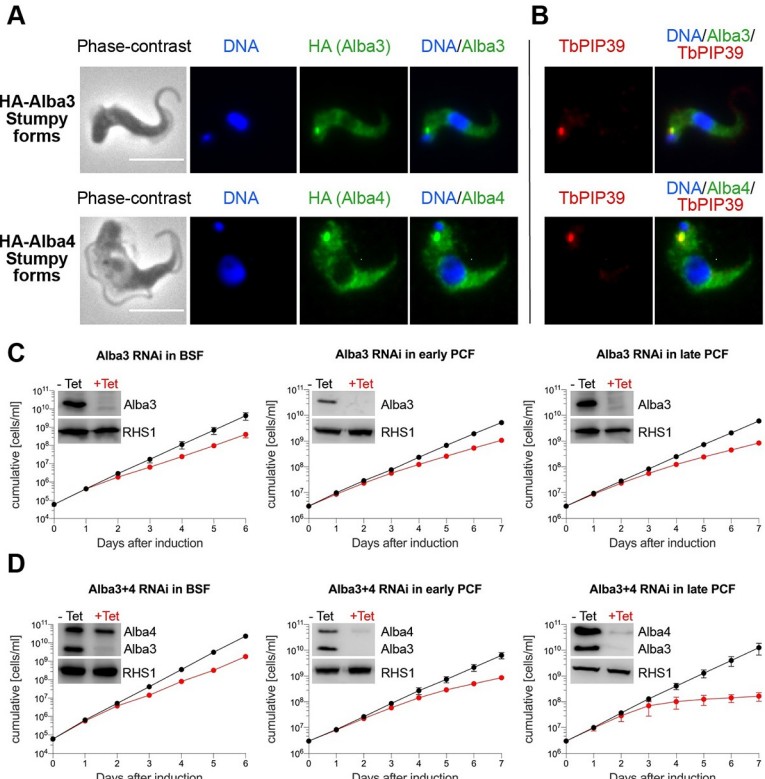

**Fig 1. Localisation of Alba 3 and Alba 4 and effects of RNAi. (A)** & **(B)**: Alba3 and Alba4 localise to the "stumpy regulatory nexus" (STuRN). Immunofluorescence analysis of HA-tagged Alba3 and Alba4. Cells were incubated with anti-HA and anti-TbPIP39. Scale bar, 5μm. **(C)** & **(D)**: Knockdown of Alba3 or Alba3+4 affects growth. RNAi was performed in slender bloodstream forms (BSF), early procyclic forms (early PCF) and late procyclic forms (late PCF). Graphs represent growth curves of induced (+Tet) and non-induced (-Tet) RNAi cell lines. Error bars, mean ±SD, n = 3. Efficiency of knockdown was assessed by Western blot analysis on day 4 after RNAi induction. RHS1 served as a loading control.

Alba3cKO showed the same growth defect as the KO (Fig 2F). When Alba3 expression was induced, the cKO grew at the same rate as AnTat 90–13 for the first 5 days, but more slowly thereafter (Fig 2F). This partial rescue at later time points was not due to over-expression, as the level of Alba3 in the cKO was comparable to AnTat 90–13 throughout the experiment (S5 Fig). Taken together, these findings suggest that Alba3 is neither essential in slender forms, nor for their differentiation to stumpy forms, but is required for differentiation to procyclic forms. In addition, Alba3 might also be essential in differentiated procyclic forms.

### *In vivo* studies of Alba3KO in mice and tsetse flies

Although Alba3 is not essential for bloodstream forms *in vitro*, it might be required *in vivo*. Groups of mice (n = 3) were infected with either AnTat 90–13 or KO slender bloodstream forms and parasitaemia was monitored at different time points. Both AnTat 90-13- and KO-infected mice exhibited similar first and second waves of parasitaemia (Fig 3A). This implies that the KO was both able to switch VSGs and to differentiate to the stumpy form. Blood smears obtained at various time points confirmed that the KO could give rise to stumpy forms *in vivo* (Fig 3A).

To assess the requirement for Alba3 in the insect vector, tsetse were infected with AnTat 90–13 or KO stumpy forms. From each group, >100 flies were dissected at days 12–15 and

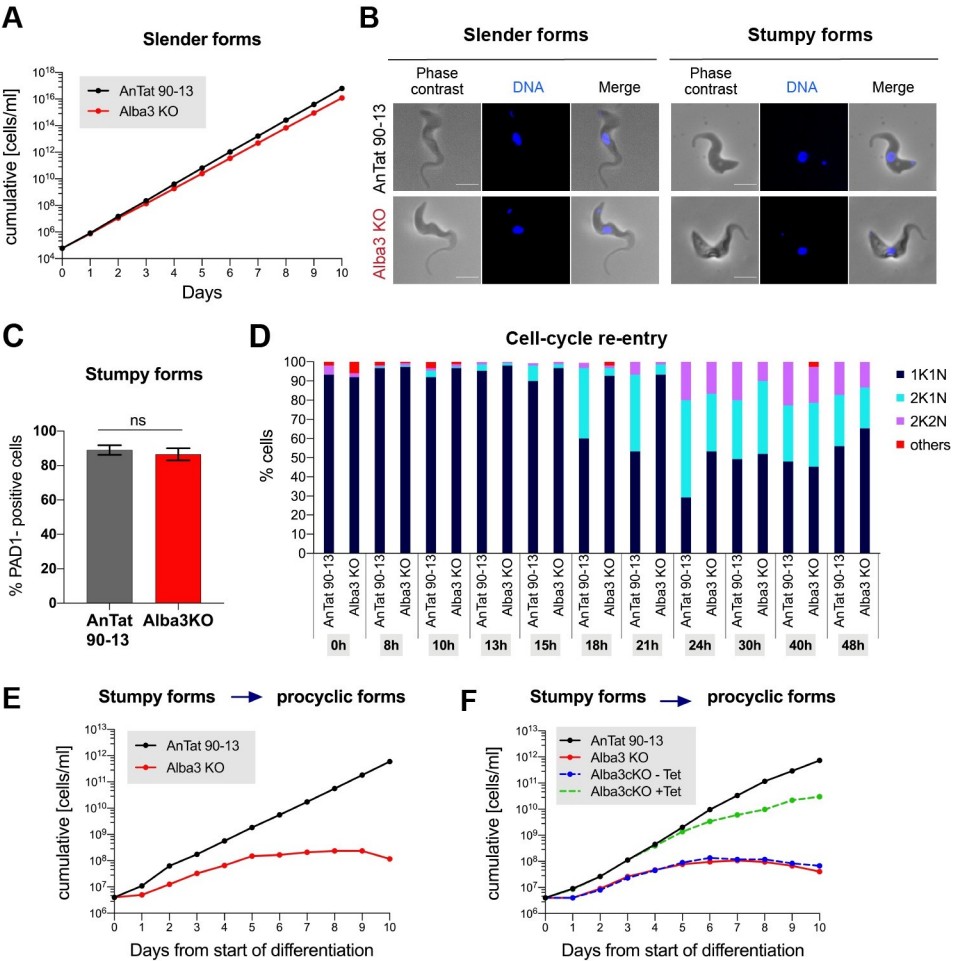

**Fig 2. Analysis of Alba3KO *in vitro*. (A)** Cumulative growth curves of AnTat 90–13 & Alba3KO bloodstream forms (n = 3). Error bars, mean ±SD. **(B)** Microscopy of AnTat 90–13 and Alba3KO slender and stumpy forms. Scale bar, 5µm. **(C)** Percentage of PAD1-positive cells. AnTat 90–13 and Alba3KO were allowed to differentiate to stumpy forms *in vitro*. An immunofluorscence assay was performed with anti-PAD1 antiserum. One hundred cells were counted per sample. Means and SD are shown. ns: not significant. **(D)** Cell cycle analysis of AnTat 90–13 and Alba3KO for the first 48h after triggering differentiation to procyclic forms. One hundred and fifty cells per sample and per timepoint were analysed for kDNA-nucleus (K-N) configuration. The figure depicts one of two biological replicates. **(E)** Cumulative growth curves of AnTat 90–13 and Alba3KO during differentiation. Cells were diluted to $4\times10^6$ ml$^{-1}$ and growth was monitored for 10 days. The graph is a representative of 3 biological replicates. **(F)** Cumulative growth curves of AnTat 90–13, Alba3KO and Alba3 conditional KO (Alba3cKO) during differentiation. Alba3 expression was induced (+Tet) or not (-Tet) in Alba3cKO throughout the differentiation starting from slender forms. Cells were counted daily and diluted to $4\times10^6$ ml$^{-1}$ if required. A representative example (one of 3 biological replicates) is shown.

assessed for the prevalence and intensity of infection in the midgut and proventriculus (Fig 3B). For flies infected with the parental line, 40% had midgut infections and 29% had proventriculus infections, whereas 18.7% of the flies infected with the KO had midgut infections and 5.4% had proventriculus infections (Fig 3B). Of note, AnTat 90-13-fed flies exhibited heavy infections in both the midgut and proventriculus, whereas KO-fed flies showed mainly weak to intermediate infections. These results suggest that, while KO parasites are capable of establishing fly infections to some extent, they are severely impaired. In an independent experiment, infections were monitored after 28–30 days (the length of time required for the parental line to establish salivary gland infections). Fig 3C shows that the midgut and proventriculus infection rates with the parental line (37% and 35%, respectively) were very similar to the earlier time

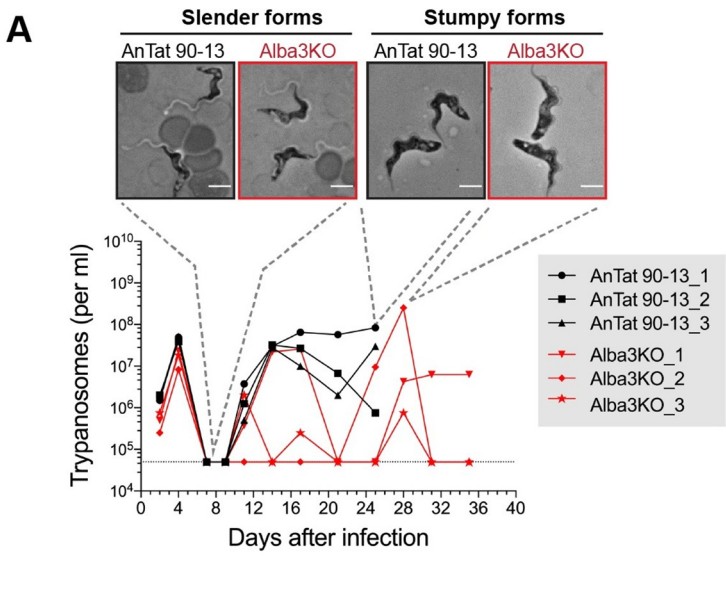

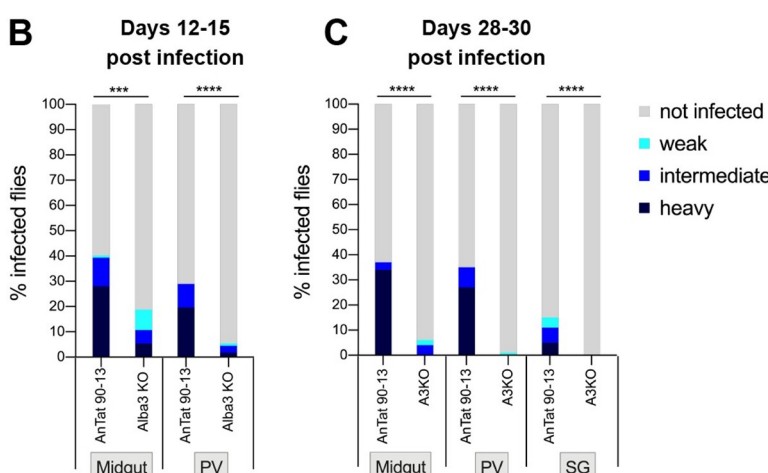

**Fig 3. Analysis of Alba3KO *in vivo*. (A)** Parasitaemia in mice infected with AnTat 90–13 or Alba3KO (n = 3 each). The upper panel shows phase contrast images of trypanosomes obtained on days 7 and 25 post infection. Mice infected with AnTat 90–13 had to be euthanised on day 25 in conformity with local regulations. **(B) & (C)** Severely impaired fly infection by Alba3KO. **(B)** Teneral tsetse flies were infected with AnTat 90–13 (n = 107) or Alba3KO stumpy forms (n = 112). Flies were dissected on days 12–15 post infection and graded for the prevalence and intensity of infections in the midgut and proventriculus. **(C)** Teneral tsetse flies were infected with AnTat 90–13 (n = 79) or Alba3KO stumpy forms (n = 83). Flies were dissected on days 28–30 post infection and graded for the prevalence and intensity of infections in the midgut, proventriculus and salivary glands. For statistical analysis total infection prevalence was compared using Fisher's exact test, two-sided. ∗∗∗$P < 0.001$, ∗∗∗∗$P < 0.0001$.

point, whereas the KO showed a reduction in midgut infections (6%, none of them heavy) and only a single weak infection of the proventriculus after 28 days. Finally,15% of flies infected with the parental line gave salivary gland infections, but none of the flies infected with the KO.

## Expression of early markers of differentiation

To gain more insights into the requirement for Alba3 during differentiation, we examined the expression kinetics of several stage-specific proteins. Stumpy forms of the parent and KO were triggered to differentiate to procyclic forms, and samples were collected over a period of 72h.

First, expression of EP and GPEET procyclins was monitored by flow cytometry. Fig 4A shows that both parent and KO parasites began to express surface EP at 4h and GPEET at 12h. The percentage of KO cells expressing EP and GPEET at these time points was slightly lower than that of the parental line, but by 24h nearly 95% of the KO population expressed both procyclins. In parallel, we monitored VSG loss during differentiation. The Western blot in Fig 4B shows that both cell lines lost VSG, but the KO was delayed by 24h. Major surface protease B (MSP-B) is another stage-specific marker [48]. Western blot analysis showed that MSP-B could first be detected at 4h in both the parent and the KO. While there was slightly less MSP-B in the KO at 4h and 6h, the levels were comparable from 8h onwards (Fig 4B). Finally, PAD1 expression was monitored [49]. Fig 4B shows that the KO exhibited a slight delay in down-regulating PAD1. In the parental line, PAD1 fell below the level of detection by day 2, but it took an additional 24h for the KO.

The relative positions of the nucleus (N) and kinetoplast DNA (kDNA) in different life-cycle stages are spatially well defined. In the stumpy form, kDNA (K) is close to the posterior end of the cell (P) and during differentiation to the procyclic form, it repositions so that it lies approximately equidistant from the nucleus and posterior end of the cell [50]. The P-K/K-N ratios of stumpy forms and procyclic forms are approximately 0.2 and 1, respectively. The distance between P-K and K-N was measured for 100 cells at each different time point during differentiation. Fig 5A shows that there is a delay in kDNA repositioning in the KO compared to the parental line, but it is completed by 48h. Taken together, these findings demonstrate that the KO is able to regulate expression of the early markers of differentiation, albeit with a slight delay relative to AnTat 90–13.

## Alba3 is required for successful development of procyclic forms

Cells deficient for Alba3 have a strong growth defect during differentiation. To investigate at which time point Alba3 is required, we used the Alba3cKO cell line. Slender forms were cultured and allowed to differentiate to stumpy forms in the absence of tetracycline, i.e. without ectopic expression of Alba3. After triggering stumpy forms to differentiate to procyclic forms, Alba3 expression was induced at different times (day 0, 2, 4, 6 or 8). Fig 5B shows that the growth defect could be rescued when Alba3 expression was induced up to 4 days from the start of differentiation, but not at later time points. Previous studies indicate that parasites commit to procyclic form differentiation within 2–3 hours after exposing stumpy forms to cis-aconitate [51,52]. After this point, cells maintain the expression of early differentiation markers and undergo onward development to become a fully differentiated procyclic form. Taking these findings into account, it seems that Alba3KO is able to respond to the differentiation trigger and initiate the differentiation process with faithful expression of early markers. However, they require Alba3 within a certain time frame for their successful progression to procyclic forms.

## Analyses of the proteomes and transcriptomes of differentiating cells

Since Alba proteins are known to be associated with the translation machinery [38], we investigated whether Alba3 had an impact on the proteome of differentiating cells. Quantitative label-free mass spectrometry performed on whole cell lysates of AnTat 90–13 and the KO on days 0 (stumpy forms), 2 and 4 of differentiation resulted in the identification of 2125, 3326 and 3447 proteins, respectively (S1 Table). These experiments revealed profound differences between the parental line and the KO (Fig 6A). On day 2, 218 proteins (6.6%) that were present in AnTat 90–13 were not detected in the KO and a further 472 proteins (14.2%) were significantly down-regulated (p value < 0.01; fold change ≥ 2) (Fig 6B). On day 4, 145 proteins

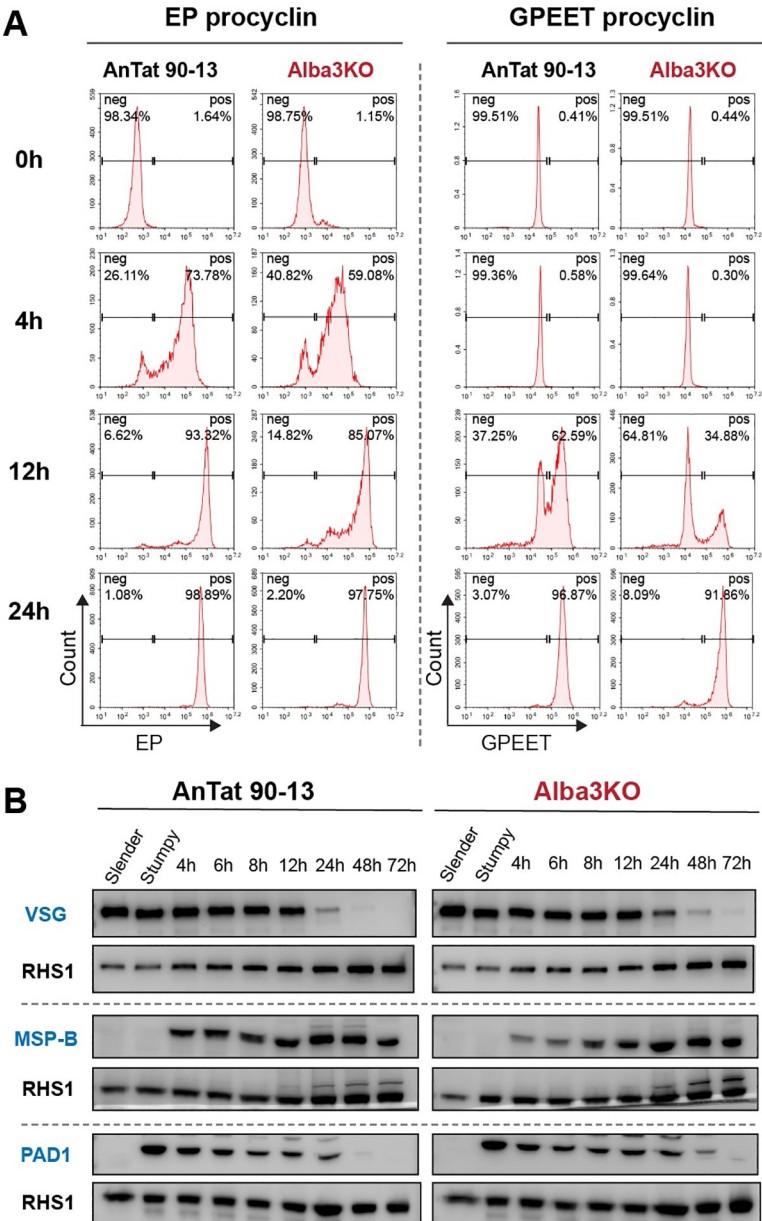

**Fig 4. Analysis of markers of differentiation from stumpy to procyclic forms. (A)** Kinetics of EP and GPEET expression in the parental line AnTat 90–13 and Alba3KO during differentiation. Cells were collected at different timepoints after exposure to cis-aconitate and a drop in temperature and flow cytometry analysis of EP and GPEET was performed. The insets indicate the percentage of cells that were positive (pos) or negative (neg). **(B)** Western blot analysis of VSG, MSP-B and PAD1 expression in AnTat 90–13 and Alba3KO during differentiation. RHS 1 served as a loading control.

(4.2%) were not detected in the KO and 277 proteins (8.1%) were significantly reduced (p value < 0.01; fold change ≥ 2) (Fig 6B). Gene Ontology (GO) term analysis on days 2 and 4, revealed significant enrichment (p value < 0.001) of proteins not detected in Alba3KO; the GO terms encompassed genes encoding mitochondrial proteins, mitoribosomal proteins and ribosomal subunits (Fig 6C). GO term analysis also revealed many enriched categories for significantly down-regulated proteins (p value < 0.01; fold change ≥ 2) in Alba3KO (S6 Fig). In

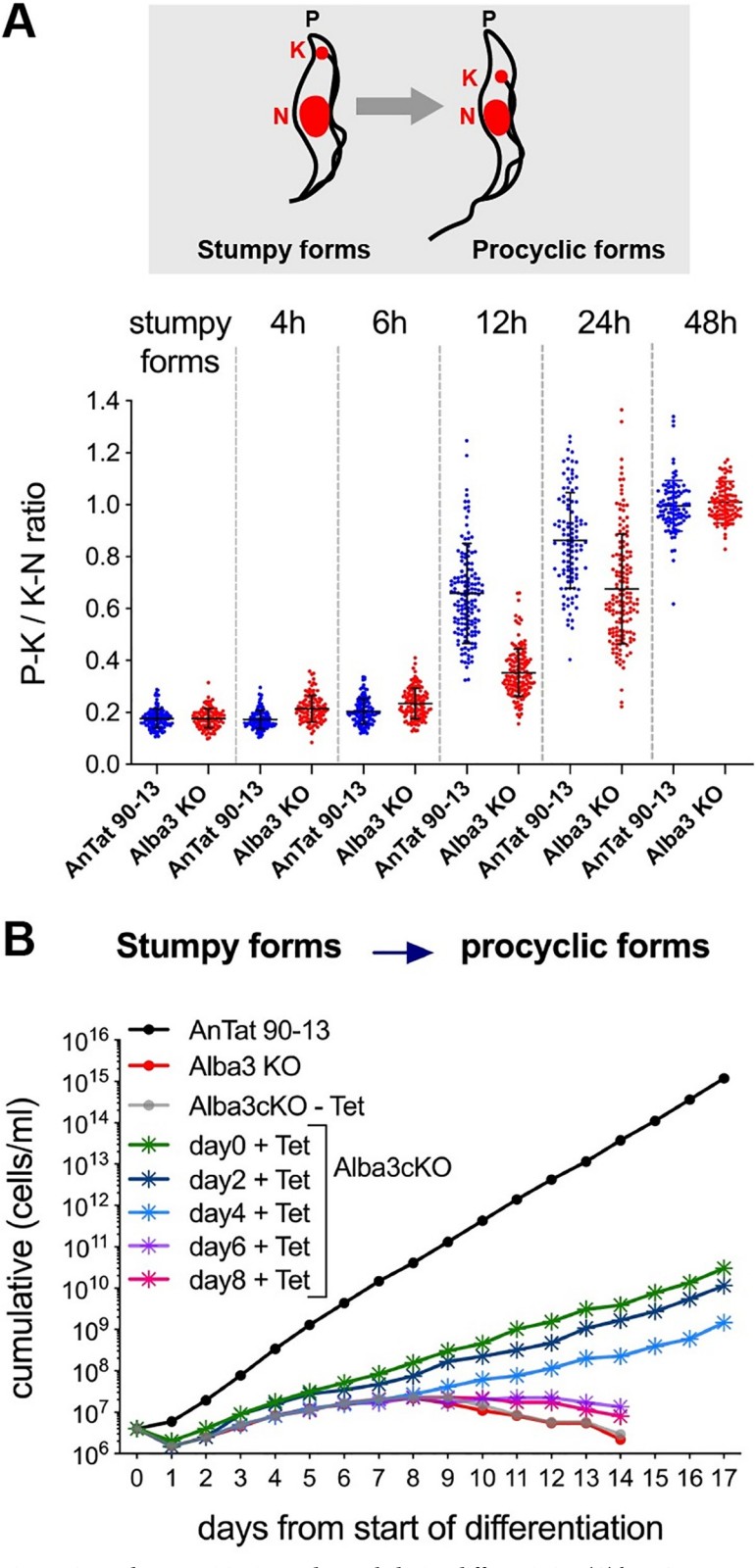

**Fig 5. Kinetoplast repositioning and growth during differentiation** (**A**) kDNA repositioning during differentiation. Top; schematic view of positions of kDNA (K) and the nucleus (N) in stumpy forms and procyclic forms. "P" indicates the posterior end of the cell. The position of the kinetoplast, relative to the nucleus (P-K/K-N) was measured in AnTat

90–13 and Alba3KO cells (n = 100) collected at different time points during differentiation. Bottom; dot plot with P-K/ P-N ratio of AnTat 90–13 and Alba3KO cells during differentiation. Means and SD are shown. **(B)** Cumulative growth curves of AnTat 90–13, Alba3KO and Alba3cKO during differentiation. Alba3 expression was induced (+Tet) at different time points in Alba3cKO. Cells were counted daily and diluted to $4x10^6$ ml$^{-1}$ if required. One of 3 representative experiments is shown.

addition, some bloodstream form-specific proteins (RDK2, POMP39-1, aldolase and alternative oxidase) were upregulated in the KO compared to AnTat 90–13 on days 2 and 4. (Fig 6D). Intriguingly, most of the proteins not detected in Alba3KO on days 2 or 4 are specific to procyclic forms, as there was very little overlap with the previously published proteome of slender forms [24] (Fig 6E). This implies that proteins that are normally newly synthesised during differentiation to procyclic forms are not produced by Alba3KO.

To determine whether the differences in protein levels reflected differences in the levels of the corresponding mRNAs, RNA-seq was performed from cells harvested at the same time points used for the proteome analyses. These showed only minor differences in mRNA levels between the parental line and the KO at all time points (Fig 7A). A comparison of mRNA expression for those proteins not detected in the KO also showed no significant differences between AnTat 90–13 and the KO (Fig 7B). There are two possible explanations for these findings: either certain proteins are not translated efficiently, or they are unstable in the absence of Alba3.

## Polysome assembly is delayed in the absence of Alba3

Since the altered protein levels in the KO were not accompanied by equivalent changes in transcript levels, we investigated whether there were differences in the loading of mRNA onto ribosomes. We first examined the polysome profiles of stumpy forms and differentiating parasites 2 and 4 days after triggering differentiation. Fig 8A shows that stumpy forms of both AnTat 90–13 and KO have virtually no polysomes. This is not unexpected, since stumpy forms are arrested in G0 and are translationally repressed [53,54]. On day 2, the profile of the parental line showed clear polysome peaks that increased further on day 4. This contrasted with the KO, which had a large peak of ribosomal subunits, but hardly any polysomes on day 2 (Fig 8A). By day 4, the polysome peaks of the KO resembled those of its parent on day 2, although the subunit and monosome peaks remained much higher (Fig 8A). These results indicate that polysomes are able to assemble during differentiation in the absence of Alba3, but with a delay of approximately 48h.

To assess association of specific mRNAs with the ribosomes, RNA was extracted from fractions from polysome gradients on days 2 and 4. The distribution of selected mRNAs was analysed by real time quantitative PCR (RT-qPCR). We selected four transcripts: COX IV, which gave 3- and 3.8-fold less protein in the KO on days 2 and 4, respectively, COX VIII, whose protein was not detected in the KO and two transcripts (EF1a and α-tubulin) whose protein and mRNA levels were unchanged. Fig 8B shows that all four mRNAs had similar distributions over the sucrose gradients, regardless of whether Alba3 was present or not. Notably, a considerable proportion of COX IV and COX VIII mRNAs were associated with polysomes from AnTat 90–13 and KO on both days. In all cases, there was increased mRNA association with the heavy polysomes on day 4 compared to day 2.

## Alba3 is required for proteome remodelling during differentiation

A previous study on the relative translation rates in slender, stumpy and procyclic forms showed that stumpy forms exhibited markedly reduced protein synthesis, as measured by

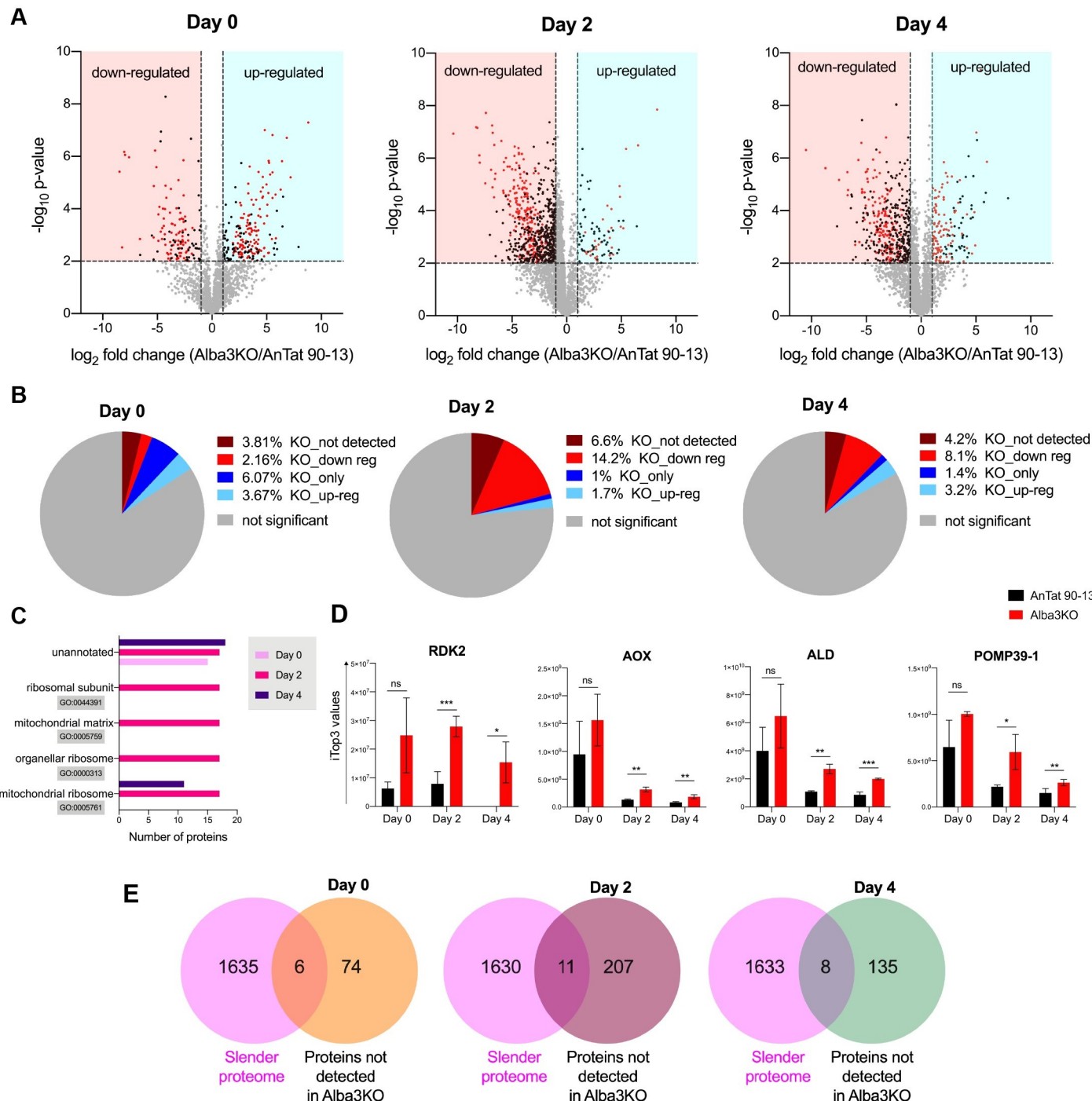

**Fig 6. Quantitative proteome analysis of Alba3KO and AnTat 90–13 during differentiation reveals significant changes.** (A) Volcano plots comparing the proteomes of AnTat 90–13 and Alba3KO on days 0 (stumpy forms), 2 and 4 of differentiation. Data were obtained from two biological replicates (with two technical replicates of each). Vertical dotted lines indicate a fold change ≥ 2 and the horizontal dotted line indicates a p value <0.01. Black dots represent differentially regulated proteins, red dots represent proteins that were detected in only one condition (i.e., detected either in AnTat 90–13 or in Alba3KO) and grey dots are proteins that do not differ significantly between AnTat 90–13 and Alba3KO. The blue region on each graph shows proteins that are up-regulated or only detected in Alba3KO. The red region shows proteins that are down-regulated or not detected in Alba3KO. (B) Pie chart indicating the percentage of differentially regulated proteins in Alba3KO on days 0, 2 and 4 of differentiation compared to AnTat 90–13. Percentages are shown for proteins that are significantly (p value <0.01, fold change ≥ 2) down/up-regulated in Alba3KO and proteins that were not detected/only detected in Alba3KO. (C) GO term analysis of significantly down-regulated proteins (p value <0.01, fold change ≥ 2) in Alba3KO compared to AnTat 90–13 on days 0, 2 and 4 of differentiation. (D) Expression levels of bloodstream form-specific proteins RDK2, POMP39-1, aldolase (ALD) and alternative oxidase (AOX) in AnTat 90–13 and Alba3KO on days 0, 2 and 4 of differentiation. Two -tailed Student's T tests were performed. Means and SD are shown. ns: not statistically signifcant; *, p = <0.05; **, p = <0.01; ***, p = <0.001. (E) Venn diagrams comparing previously published slender proteome [24] proteins not detected in the proteome of Alba3KO on days 0, 2 and 4.

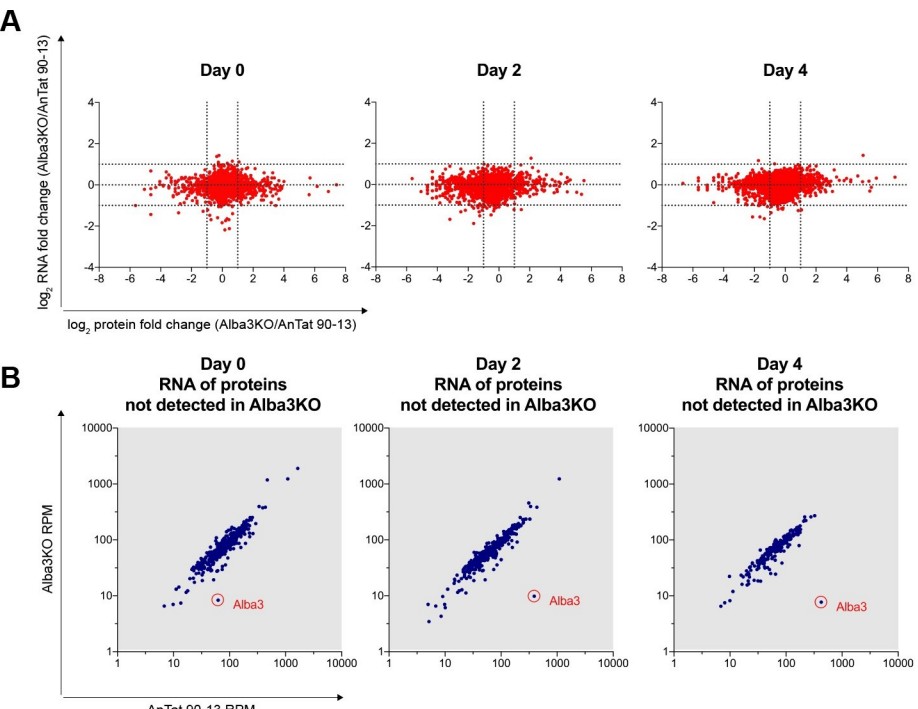

**Fig 7. Differences in proteomes do not reflect the transcriptomes.** (**A**) Comparison of the log$_2$-fold changes between the proteomes and transcriptomes of AnTat 90–13 and Alba3KO on days 0, 2 and 4 of differentiation. Transcriptome data were obtained from three biological replicates. The dotted lines indicate a fold change of $\geq 2$. (**B**) Scatter plot comparing the transcript levels (RPM, reads per million) in AnTat 90–13 and Alba3KO of proteins that were not detected in Alba3KO.

incorporation of $^{35}$S-methionine, but escaped from translational repression during differentiation to procyclic forms [54]. Since mRNAs were still loaded onto polysomes in the absence of Alba3, we monitored protein synthesis in the presence of the methionine analogue L-homo-propargylglycine (L-HPG). Newly synthesised proteins containing L-HPG were covalently coupled to azide-biotin, purified using streptavidin beads and used for Western blot analysis. Global translation was monitored on days 2 and 4 in AnTat 90–13 and Alba3cKO in which ectopic Alba3 expression was induced (+Tet) or not (-Tet). Detection with streptavidin-HRP (Fig 8C) indicates that proteins were efficiently synthesised in the AnTat 90–13 and Alba3cKO +Tet samples on day 2, whereas labelled proteins were not detected in Alba3cKO -Tet. On day 4, however, there were similar levels of newly synthesized proteins in Alba3cKO irrespective of whether Alba3 was expressed or not (Fig 8C). To monitor the synthesis of individual proteins we made use of antibodies for COX IV and α-tubulin. The lower panel of Fig 8C shows that newly synthesised COX IV is only weakly detected on day 2 in the absence of Alba3 (-Tet). By contrast, the level of newly synthesised tubulin was less affected, although slightly reduced. By day 4, both COX IV and tubulin were made at comparable levels, irrespective of whether Alba3 is expressed or not.

The results shown above suggest that Alba3 assists escape from translational repression during differentiation. It was shown previously in fully differentiated procyclic forms that Alba3 is associated with ribosomal proteins, poly(A)-binding proteins and the translation initiation factor eIF4E4 [38]. It was also shown that the bulk of Alba3 cosedimented with ribosomal subunits and monosomes, with a smaller amount detectable in the polysome fractions. To determine whether Alba3 exhibited the same profile during differentiation, we performed

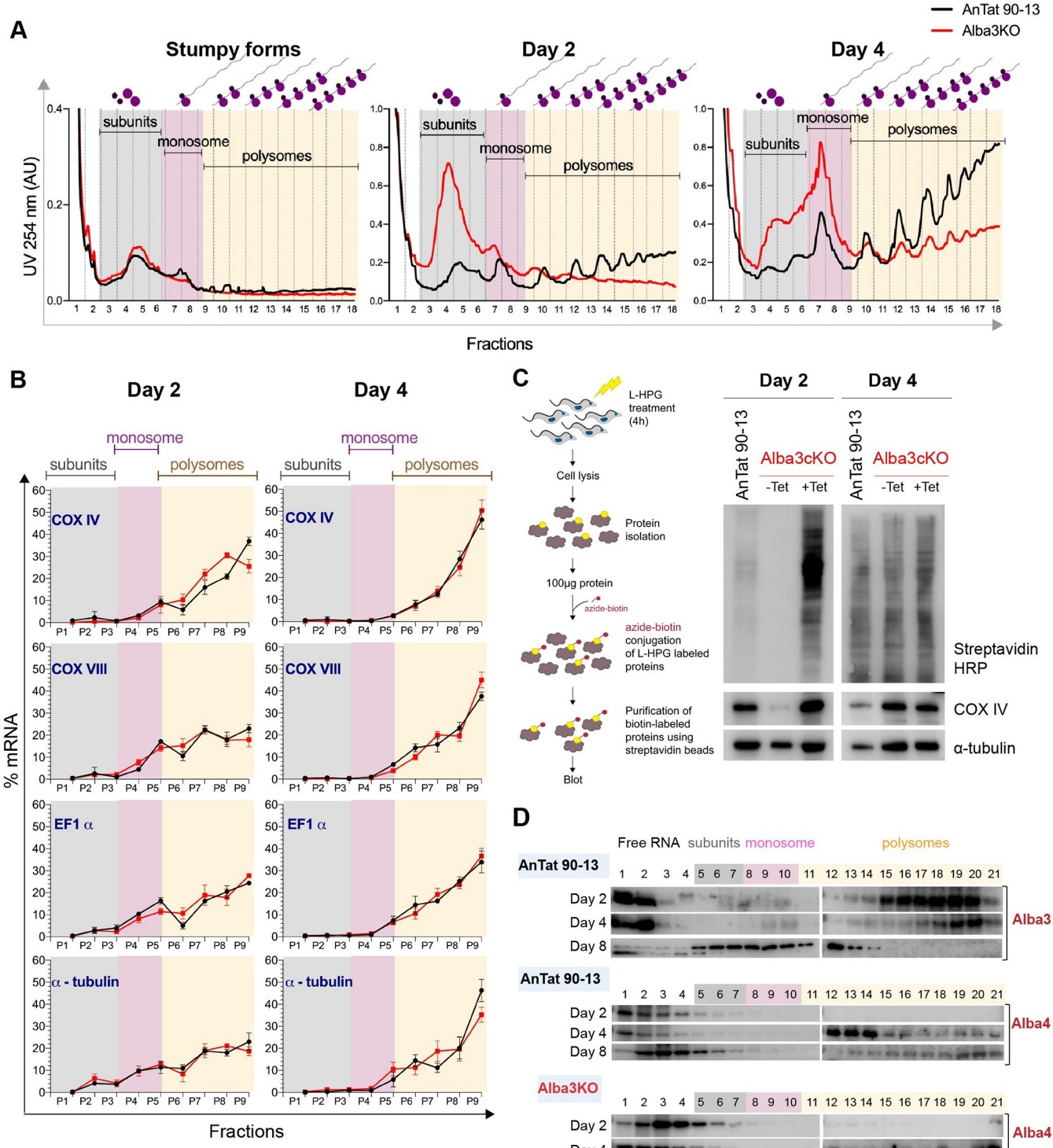

**Fig 8. Polysome profiles and translation during differentiation. (A)** Polysome profiles of AnTat 90–13 and Alba3KO cells harvested at day 0 (stumpy forms), day 2 and 4 of differentiation. Cell lysates were overlaid on 10–50% sucrose gradients and subjected to ultracentrifugation. Gradients were collected from top (fraction 1) to bottom (fraction 18) while continuously monitoring UV absorbance at 254nm to determine the concentration of RNA. For technical reasons (see Materials and Methods) 3-fold fewer cell equivalents were loaded for stumpy forms of both cell lines. The figure is a representative example of one of three biological replicates. **(B)** Distribution of mRNAs for COX IV, COX VIII, EF1a and α-tubulin determined by RT-qPCR. Total RNA was extracted from the sucrose gradient fractions in **(A)**. Pairs of consecutive fractions were pooled (e.g. fractions 1 and 2) giving rise to RNA fractions P1-P9. The distribution of mRNA is shown as the percentage of total mRNA (% mRNA) in each fraction. Error bars, mean ±SD of 3 technical replicates. **(C)** Analysis of newly synthesised

proteins in AnTat 90–13 and Alba3cKO on day 2 and day 4 of differentiation. Ectopic Alba3 expression was induced (+Tet) or not (-Tet) by addition of tetracycline. Left; Diagram illustrating the experimental procedure. Right; Western blot analysis. Newly synthesised proteins were detected using streptavidin-HRP. Newly synthesised COX IV and α-tubulin proteins were detected using specific antibodies. **(D)** Distribution of Alba3 and Alba4 proteins over the sucrose gradients. The upper and middle panels represent the Western blot analyses of Alba3 and Alba4, respectively, in sucrose gradient fractions obtained from the parental line AnTat 90–13 on days 2, 4 and 8 of differentiation. The lower panel represents Western blot analyses of Alba4 in fractions obtained from Alba3KO on days 2 and 4. The condition of these cells by day 8 precluded any analyses.

Western blots with fractions obtained from the sucrose gradients. Fig 8D shows that on day 2, Alba3 was mainly present in the polysome and free RNA fractions, whereas on day 4, the amount of Alba3 in the polysome fractions was reduced. By day 8, at which point the trypanosomes can be considered fully differentiated, the majority of Alba3 was detected in the ribosomal subunit and monosome fractions, together with a small amount in the early polysome fractions. This implies that Alba3 has a specific role in translation during the transition between life-cycle stages.

## Partial redundancy of Alba3 and Alba4

As mentioned above, Alba3 and Alba4 are closely related to each other. To test if there is functional redundancy between them, we made a knockout of Alba4 (Alba4 KO) in slender bloodstream forms (S4D and S4E Fig). As shown in Fig 9A, the Alba4KO grew slightly more slowly (population doubling time 6.5h ±0.08h) than the parental line (6h, ±0.10h), but the difference was not statistically significant (p = 0.079). Alba4 KO slender forms were able to differentiate to stumpy forms. Interestingly, upon differentiation to procyclic forms, Alba4KO grew with the same kinetics as its parent for the first 4 days; after that it still grew exponentially, but more slowly (Fig 9B), This is in stark contrast to the slow growth exhibited by the Alba3KO from day 1, followed by severe growth arrest from day 5 onwards.

Since Alba3 and Alba4 genes lie adjacent to each other on chromosome 4, we attempted to knock them out simultaneously. It was possible to knockout one allele of Alba3+4 in slender forms by homologous recombination, but we were not able to knock out the second allele. Because of this, we transfected the Alba3+4 single KO with a tetracycline-inducible ectopic copy of Alba3. When expression of Alba3 was induced, it was possible to delete the remaining endogenous copies of Alba3+4 (A3&4KO+iA3). This was confirmed by PCR (S4D Fig). Slender forms of A3&4KO+iA3 grew normally as long as tetracycline was present, but displayed a drastic growth defect 3 days after it was removed (Fig 9C). These results suggest that there is functional redundancy between Alba3 and Alba4 in slender forms. Moreover, differentiation of A3&4KO+iA3 from stumpy forms to procyclic forms revealed that, upon tetracycline removal on days 0, 2, 4, 6 or 8, parasites were neither able to survive the differentiation process nor survive as procyclic forms (Fig 9D).

As mentioned above, we observed that Alba3 predominantly cosedimented with polysomes on day 2 of differentiation, whereas, on day 8 it was present in the ribosome subunit fractions and monosome fractions. When the distribution of Alba4 protein was analysed, it showed a different profile. Fig 8D shows that on day 2, Alba4 is present only in the free RNA fractions, with very little in the subunit fractions and none in the polysome fractions. By day 4, a large amount of Alba4 was present in the early polysome fractions and very little in the late polysome fractions. By day 8, Alba4 was once again detected in the free RNA and subunit fractions, with very little in the polysome fractions. In summary, these results suggest that while there is functional redundancy between Alba3 and Alba4 in slender forms, Alba4 cannot substitute for Alba3 during differentiation to procyclic forms.

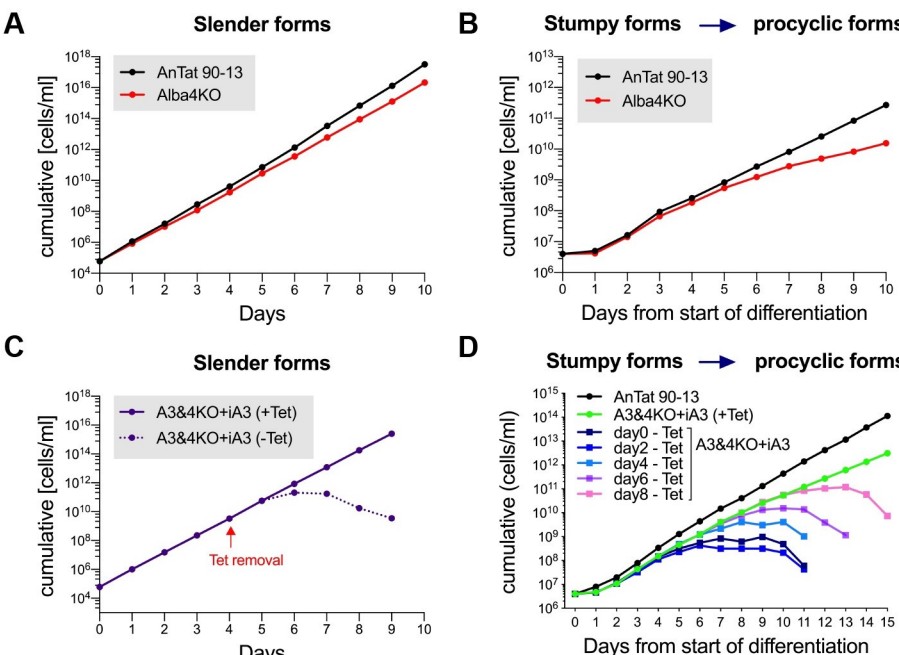

**Fig 9. Partial redundancy of Alba3 and Alba4. (A)** Cumulative growth curves of AnTat 90–13 & Alba4KO bloodstream forms (n = 3). Error bars, mean ±SD. **(B)** Cumulative growth curves of AnTat 90–13 and Alba4KO during differentiation. Cells were counted daily and diluted to $4x10^6$ ml$^{-1}$ if required. **(C)** Cumulative growth curves of Alba3&4 knockout + inducible Alba3 expression cell line (A3&4KO+iA3). Growth of A3&4KO+iA3 bloodstream forms was monitored in the presence of tetracycline (+Tet; induces Alba3 expression) or upon removal of tetracycline (-Tet). **(D)** Cumulative growth curves of A3&4KO+iA3 cells during differentiation. A3&4KO+iA3 slender forms were cultured and allowed to differentiate to stumpy forms in the presence of tetracycline (+Tet). Thereafter, tetracycline was removed at different time points from the start of differentiation. Cells were counted daily and diluted to $4x10^6$ ml$^{-1}$ if required.

## Discussion

Over the course of evolution, Alba-domain proteins have exhibited great diversity in their functions. In this study we provide evidence that *T. brucei* Alba3 plays an important role in proteome remodelling during differentiation from the mammalian stumpy form to the insect procyclic form.

The four Alba proteins in *T. brucei* are present in thousands of copies per cell [38], suggesting a broad requirement for them. Nevertheless, depletion of Alba1, 2 or 4 by RNAi had no discernible effect on growth of bloodstream forms and procyclic forms and knock down of Alba3 had only a modest effect ([38] and this study). Knockdown of Alba3+4 together resulted in a stronger growth defect in late procyclic forms, but did not show an additive effect in bloodstream forms. It is worth noting, however, that in contrast to what was observed for procyclic forms ([38] and this study) Alba4 escaped knock down by RNAi in bloodstream forms. This was not due to the cell line, because the same result was obtained with bloodstream forms derived after tsetse transmission of the original cell line generated by Mani and coworkers [38]. Individually, Alba3 and Alba4 are not essential in bloodstream forms. We were unable to delete both genes simultaneously, however, unless an ectopic inducible copy of Alba3 was present. Thus, there is functional redundancy of the two proteins in slender bloodstream forms, which is compatible with their high degree of identity. Slender forms of Alba3KO retained the ability to develop into stumpy forms and express PAD1 in culture, and also gave rise to stumpy forms at peak parasitaemia in mice. It is also likely that they underwent antigenic variation, as

infections persisted for several weeks in immunocompetent mice. In contrast, the KO was compromised in establishing infections of the tsetse midgut and proventriculus, which pointed to a requirement for Alba3 in differentiation from stumpy to procyclic forms, growth as procyclic forms and/or development into later stages in the life cycle. A minority of flies became infected with the KO and sustained infections for up to 30 days. This most probably reflects heterogeneity in the fly population, with some being more permissive to infection than others. It does not imply that there is selection for a compensatory mutation in the KO. It was observed previously that trypanosomes lacking procyclin genes infected the salivary glands at a lower rate than wild-type parasites. When the mutant parasites were passaged through mice, however, and used to re-infect flies, there was no increase in their salivary gland infectivity [55].

*In vitro*, Alba3KO was able to initiate the differentiation program to procyclic forms (expression of procyclins and MSP-B, repositioning of kDNA), although with slightly slower kinetics than its parent, suggesting that Alba3 was not essential for these early events. Alba3 was indispensable at later time points, however, as the KO had a longer population doubling time and stopped dividing within 5–6 days. It therefore seems that differentiation can be separated into Alba3-independent and -dependent processes. In support of this, Alba3cKO was able to rescue the growth defect when Alba3 expression was induced up to 4 days after triggering differentiation of stumpy to procyclic forms. Beyond this time, cells underwent the same growth arrest as Alba3KO between days 5 and 6 and did not resume division. Alba4 is not required for differentiation, but may compensate for the lack of Alba3 in the first 24–48 h, since the two proteins are extremely similar and may associate with the same core set of factors. The specific need for Alba3 beyond this time might reflect interactions that are mediated by the unique regions and/or post-translational modifications of the protein.

Deletion of Alba3 had little effect on steady state mRNA levels in stumpy forms and differentiating parasites, but major effects on the proteome. On day 2 after inducing differentiation, 218 proteins were not detected in Alba3KO and 145 were not detected on day 4. The majority of proteins not detected in the KO were developmentally regulated proteins such as COX subunits, components of the mitoribosome and other mitochondrial proteins. Several hundred proteins that were expressed at lower levels in the knockout were also specific for procyclic forms (72% on day 2 and 81% on day 4). Conversely, some bloodstream form-specific proteins persisted longer in Alba3KO. Taken together, this indicates a requirement for Alba3 in order for the trypanosomes to remodel their proteome during differentiation. Mitochondrial maturation, in particular, is key to the metabolic switch from glucose to proline as an energy source [56,57]. In addition, Alba3 could be required for maintenance of the procyclic proteome and for further development. In support of this, the double knockdown of Alba3+4 by RNAi gave a stronger growth defect in late procyclic forms. Alba3 itself is also under developmental control: it is down-regulated during the transition from mesocyclic to epimastigote forms [39], which may be another point at which trypanosomes remodel their proteome as they adapt to the proventriculus.

Stumpy forms are translationally repressed and protein synthesis first resumes when the parasites undergo differentiation [53,54]. Consistent with this, ribosomal subunits and monomers, but no polysomes, were detected in stumpy forms of both the parental line and the KO. In the case of AnTat 90–13, polysomes were clearly seen by day 2, while the KO first showed distinct polysome peaks on day 4. Might depletion of Alba3 lead to a defect in ribosome biogenesis? This seems unlikely, as Alba3 is never localised to the nucleolus, which is the site of ribosome biogenesis. Moreover, the KO was still able to assemble ribosomal subunits and monosomes, which formed the dominant peaks for this cell line on day 2 and day 4, respectively. It is possible that not all of them were functional, however. On days 2 and 4 post

differentiation Alba3 cosediments with polysomes of AnTat 90–13, whereas by day 8 (this study) or in fully differentiated procyclic forms [38] it is mainly in the subunit and monosome fractions. Thus, Alba3 might play a role in polysome formation during the stumpy to procyclic transition. Alternatively, the effect might be indirect since the Alba3KO produces less TbL11, an essential factor in ribosome maturation [58], than the parental cells. In either case, however, it would imply that only a subset of translating ribosomes are affected, since the majority of proteins are present at similar levels in both cell lines. Specialised ribosomes have been described in a variety of organisms [59–62], but have not yet been reported for trypanosomes. Interestingly, Alba4 was mainly present in the free RNA fractions of both AnTat 90–13 and KO cells on day 2, and then shifted progressively to the early polysomes on day 4 and the late polysomes on day 8, implying that its main role is in fully differentiated cells. Stage-specific cosedimentation with polysomes has previously been documented for the zinc finger protein ZFP3 [63]. Once again, it is not known whether ZFP3 associates with all ribosomes or whether *T. brucei* possesses different sub-populations of ribosomes that translate different mRNAs. As noted above, there appears to be an Alba3-dependent phase of differentiation. One possibility is that Alba3 is required for the pioneer round of translation [64]; this could explain why proteins that are normally first expressed in procyclic forms are most affected in the KO. Once this has occurred, and the mature ribonucleoprotein complex is formed, there would no longer be an absolute requirement for Alba3 in subsequent rounds of translation. Yet another possibility is that Alba3 influences the speed of translation. This could explain why there are reduced levels of certain proteins despite similar sedimentation profiles of their mRNAs in the parental line and Alba3KO.

Finally, we speculate that Alba-domain proteins might play similar roles in proteome remodelling in other protozoan parasites. This may be particularly relevant under circumstances which require adaptation to sudden changes in environment, such as the transition between hosts. Since Alba-domain proteins in protozoa are best known for their RNA-binding functions, previous studies have tended to focus on the transcriptome and interacting RNAs, without examining global changes in the proteome [40,41,43–46,65,66].

## Materials and methods

### Ethics statement

Mouse infections were carried out at the Swiss Tropical and Public Health Institute (Basel) according to the rules and regulations for the protection of animal rights ("Tierschutzverordnung") of the Swiss Food Safety and Veterinary Office. Approval was granted by the Veterinary Office of the Canton Basel-Stadt, Switzerland (Licence number 2374).

### Raw data

The underlying numerical data for Figure panels 1C, 1D, 2A, 2C, 2D, 2E, 2F, 3A, 3B, 3C, 5A, 5B, 6A, 6C, 6D, 7A, 7B, 8A, 8B, 9A, 9B, 9C, 9D and S3 and S6 Figs are provided in supplementary data sheets (S1 Data).

### Cell culture

*Trypanosoma brucei brucei* EATRO 1125, clone AnTat 1.1 [67] and its derivative AnTat 90–13 [68] were used in this study.

Pleomorphic bloodstream forms were cultured in HMI-9 [69] supplemented with 15% horse serum (HS) or HMI-9 supplemented with 10% foetal bovine serum (FBS) and 1.1% methyl cellulose (MC) [70] at 37°C and 5% $CO_2$. Transfections, growth curves and RNAi

experiments with bloodstream forms were carried out in HMI-9+15% HS. To obtain a synchronous population of stumpy forms for differentiation experiments, cells were grown in HMI-9 supplemented with 10% FBS and 1.1% MC.

Early procyclic forms were cultured in DTM [71] supplemented with 15% FBS at 27˚C and 2.5% $CO_2$. Late procyclic forms were cultured in SDM79 [72] supplemented with 10% FBS at 27˚C.

## Trypanosome stocks and constructs

*T. b. brucei* AnTat 1.1 [67] was used for generating in situ HA-tagged cell lines and AnTat 90–13 [68] for RNAi cell lines and knockout cell lines. Primers, plasmids and restriction sites used for cloning constructs and the restriction sites used to digest plasmids for transfection are listed in S2 Table. Genomic DNA of *T. b. brucei* AnTat 1.1 served as a template for PCR amplification of inserts for cloning.

**RNAi constructs.** Alba RNAi constructs in the stem-loop vector pSLComp1 used in this study were described previously [38].

***In situ* N-terminal HA-tagged constructs.** In situ HA-tagging of Alba1, Alba2, Alba3 and Alba4 was performed by amplifying the first 317bp, 180bp, 243bp and 284bp, respectively, of their coding regions (CDS) excluding the start codon, and inserting them into pN-PURO-HA, a derivative of pC-PTP-NEO [73].

**Knockout constructs.** To generate Alba3 knockout (KO) constructs, 607bp upstream and 598bp downstream of the Alba3 CDS were amplified by PCR. The fragments were then cloned sequentially into PDEB1_Puro_KO (puromycin resistance) and PDEB1_Blast_KO (blasticidin resistance) [74] replacing 5' and 3' flanking regions of these constructs to generate Alba3_Puro_KO and Alba3_Blast _KO constructs. Similarly, for Alba4 KO, 608bp upstream and 638bp downstream of the Alba4 CDS were amplified by PCR. The fragments were then cloned sequentially into PDEB1_ KO vectors replacing 5' and 3' flanking regions of these constructs to generate Alba4_Puro_KO and Alba4_Blast _KO constructs. For Alba3+4 KO, 607bp upstream of the Alba3 CDS and 638bp downstream of the Alba4 CDS were amplified by PCR. These fragments were cloned sequentially into the PDEB1_KO vectors replacing 5' and 3' flanking regions of these constructs to generate Alba3+4_Puro_KO and Alba3+4_Blast_KO constructs.

**Inducible expression constructs.** For inducible expression of Alba3, the Alba3 CDS was amplified and cloned into pLEW100 [75].

## Transfection of trypanosomes

Ten micrograms of appropriately digested plasmid were used for transfection. S2 Table lists the restriction site(s) used for digestion.

Bloodstream forms: slender forms were cultured in HMI-9 + 15% HS. Forty million cells derived from a log phase culture were resuspended in 100µl of TbBSF transfection buffer [76] containing 10µg plasmid DNA and subjected to nucleofection using the program Z-001 (Amaxa Nucleofector 2b device, Lonza, Switzerland). Following this, cells were immediately transferred to 30ml of HMI-9 + 15% HS and distributed into 24-well plates at dilutions of 1:50, 1:100 and 1:500. Twenty hours post transfection, stable transformants were selected with the appropriate antibiotics (2.5 µg/ml hygromycin, 5 µg/ml blasticidin, 0.5 µg/ml puromycin or 1.5 µg/ml phleomycin). Clones were transferred to HMI-9 supplemented with 10% FBS and 1.1% MC and henceforth cultured and maintained in the same medium.

Stable transfections of procyclic forms were performed as described previously [77] and clones were selected using antibiotics at concentrations of 25 µg/ml hygromycin, 10 µg/ml blasticidin, 1µg/ml puromycin or 2.5 µg/ml phleomycin.

### *In vitro* differentiation

The slender bloodstream forms used in this study were cultured in HMI-9 supplemented with 10% FBS and 1.1% MC by maintaining them at densities <$10^6$ cells /ml at 37˚C. Differentiation to stumpy forms was achieved by allowing the cells to reach densities >$5x10^6$ cells/ml and monitoring them microscopically for stumpy cell morphology. Cultures with ~90% stumpy forms were used for differentiation to procyclic forms. Differentiation to early procyclic forms was triggered by resuspending the stumpy forms in DTM + 15% FBS [71] containing 6mM *cis*-aconitate [72] at a density of $4x10^6$ cells/ml and incubation at 27˚C. Twenty four hours after the differentiation trigger, cells were diluted daily to $4x10^6$ cells/ml in DTM + 15% FBS at 27˚C. Samples were collected for flow cytometry, microscopy and western blot analysis at different time points through and after differentiation.

### *In vivo* experiments

Mouse infections per group, 3 NMRI female mice (Charles River, Sulzfeld, Germany) were infected with long slender bloodstream form trypanosomes by intraperitoneal injection of 2 x $10^4$ parasites per mouse. Parasitaemia was monitored for 4 weeks, three times a week, by counting parasites from tail blood samples. Five microlitres tail blood was mixed with 20μl of 3.2% sodium citrate, and 5μl of the dilution was transferred to a haemocytometer. Parasites were then counted under a bright field microscope with 200x magnification. For slender/stumpy detection, blood smears were made on microscope slides, fixed with methanol and subsequently stained with Giemsa stain [78].

**Tsetse fly infection.** Pupae of *Glossina morsitans morsitans* were obtained from the Department of Entomology, Slovak Academy of Science, Bratislava. Teneral flies were infected with 2 x $10^6$ /ml freshly differentiated stumpy forms in bovine blood (Department of Entomology, Slovak Academy of Science, Bratislava). Infections and maintenance were performed as described [79], except that bovine blood was used instead of horse blood. Twelve to fifteen days or 28–30 days post infection, midguts, proventriculi and salivary glands were dissected and monitored for the presence of trypanosomes. Infections were graded as previously described [79].

### Flow cytometry

To monitor the proportion of the population expressing EP and/or GPEET procyclins, flow cytometry was performed as described previously [19,20]. The primary antibodies were anti-EP (mouse monoclonal TBRP1/247, Cedarlane Laboratories, Burlington, Canada) and anti-GPEET (rabbit polyclonal K1) [79] and the secondary antibodies were Alexa Fluor 488-conjugated goat anti-rabbit (Invitrogen, Eugene, Oregon, USA) and Cy3-conjugated goat anti-mouse (Jackson ImmunoResearch). Antibody dilutions are mentioned in S3 Table. Fluorescence of 20,000 cells was measured using a Novocyte flow cytometer system (ACEA Biosciences, Inc., San Diego, USA).

### Western blot analysis

For most experiments total cell protein extracts were prepared by resuspending cells in Laemmli buffer (CSH protocols) followed by heating at 95˚C for 5 min. For PAD1 detection, cells were resuspended in Laemmli buffer followed by sonication to shear genomic DNA [49]. Protein samples ($10^5$ cell equivalents per lane for VSG, $2x10^6$ cell equivalents per lane for all others) were then resolved on SDS polyacrylamide gels and blotted onto PVDF Immobilon-P membrane (Merck Millipore, Tullagreen, Carrigtwohill, Ireland). The membranes were

blocked in 1%BSA/TBS-Tween for 2h at room temperature and incubated overnight with primary antibody at 4°C. Following three washes with TBS/Tween, the membranes were incubated with HRP conjugated secondary antibody for 2h at room temperature. After three washes with TBS/Tween, the membranes were visualized using SuperSignal West Pico PLUS Chemiluminescent substrate (ThermoFischer Scientific, Waltham, MA, USA) using an Amersham Imager 600 (GE Healthcare, Marlborough, MA, USA).

The following primary antibodies were used: Affinity-puriflied rabbit polyclonal anti-Alba1, anti-Alba2, anti-Alba3 and anti-Alba4 [38]; sheep anti-MSP-B [80] and anti-COX IV (a kind gift from Andre Schneider, University of Bern); rabbit anti-PAD1 (a kind gift from Keith Matthews, University of Edinburgh); anti-VSG AnTat 1.1 (a kind gift from Jay Bangs, University of Buffalo), rabbit anti-RHS1 [81] and mouse monoclonal anti-α-tubulin (Sigma Aldrich, St. Louis, Missouri, USA).

The following secondary antibodies were used: swine anti-rabbit HRP (Dako Denmark); donkey anti-rat HRP (Invitrogen, Eugene, Oregon, USA); anti-mouse HRP (Dako Denmark) and anti-sheep HRP (Invitrogen, Eugene, Oregon, USA).

Antibody dilutions are mentioned in S3 Table.

## Immunofluorescence staining and microscopy

Immunfluorescence staining was carried out as described [82]. Phase contrast and immunofluorescence microscopy images were captured using Leica DM5500 B and analysed using Fiji [83].

Primary antibodies used for immunostaining were: rat anti-HA 3F10 (Roche); rabbit anti-TbPIP39 and anti-PAD1 (from Keith Matthews, University of Edinburgh). Secondary antibodies used were: Alexa Fluor 488 donkey anti-rat (Life Technologies, Eugene, Oregon, USA); Alexa Fluor 488 goat anti-rabbit and cy3 goat anti-mouse (Invitrogen, Eugene, Oregon, USA). Antibody dilutions are mentioned in S3 Table.

To monitor kDNA repositioning during differentiation cells were harvested by centrifugation, washed once with PBS and fixed in 4% paraformaldehyde overnight at 4°C. Cells were then washed once with PBS and mounted with VectaShield containing 4, 6-diamidino-2-phenylindole (DAPI) (Vector Laboratories Inc., Burlingame, California, USA). Phase contrast and DAPI stained fluorescent images were captured using a Leica DM5500 B and images were superimposed using Fiji [83]. These superimposed images were used to measure the distances between (i) the posterior end and kDNA and (ii) the distance between the kDNA and centre of nucleus using Fiji analysis software [83].

## Cell cycle analysis

To monitor cell-cycle re-entry during differentiation, $2 \times 10^6$ cells were harvested by centrifugation, washed once with PBS and fixed in 4% paraformaldehyde overnight at 4°C. Cells were then washed once with PBS and mounted with Mowiol containing Hoechst dye (10mg ml$^{-1}$). Phase contrast and Hoechst stained fluorescent images were captured using a Leica DM5500 B and images were superimposed using Fiji [83]. 150 cells were counted per sample and per time point.

## Polysome profiling

Approximately $5 \times 10^9$ cells were treated with 100μg/ml cycloheximide for 10 min to stabilise the interaction between mRNA and ribosomes prior to harvesting. Cells were pelleted by centrifugation (600g, 10 min), and washed twice with cold PBS, supplemented with 100μg/ml cycloheximide. The pellet was resuspended in 1ml cold lysis buffer (10mM NaCl, 10mM

$MgCl_2$, 10mM Tris-HCl pH7.5, 1% Triton X-100, 1% sodium deoxycholate, 0.2U/μl RNase inhibitor, 1mM DTT, 100μg/ml cycloheximide) and immediately placed on ice for 2 min. The lysate was then passed 10 times through a 27G needle, followed by centrifugation (16,000g, 10 min) at 4˚C to pellet nuclear/cellular debris, and the resulting supernatant was flash frozen in liquid nitrogen and stored at -80˚C until use. To ensure equal loading of samples, the absorbance ($OD_{A260}$) of samples was measured and the same amount of lysate (equivalent $OD_{A260}$) was loaded carefully on top of a 15–50% sucrose gradient prepared in polysome buffer (30mM Tris-HCl pH 7.5, 100mM NaCl, 10mM $MgCl_2$, 100μg/ml cycloheximide) in poly clear ultracentrifuge tubes (Science services, Munich, Germany). Owing to the presence of methylcellulose in the culture medium for bloodstream forms (which required resuspension of the cells in larger volumes), 6 $OD_{A260}$ cell equivalents were loaded for stumpy forms and 17 $OD_{A260}$ were loaded for samples from days 2 and 4. Samples loaded onto the gradients were subjected to ultracentrifugation (40,000 rpm, 2.5h) at 4˚C using a Beckman Coulter SW 41 Ti swing bucket rotor in a Beckman Coulter Optima XPN-80 Ultracentrifuge. After centrifugation, 500μl fractions were collected. UV absorption (254nm) was measured continuously using a Piston Gradient Fractionator (BioComp, New Brunswick, Canada).

## RNA isolation from polysome fractions

RNA from each 500 μl fraction was extracted with an equal volume of acid phenol-chloroform and precipitated overnight with one volume of isopropanol and 10 μg glycogen at -20˚C. After centrifugation for 1 h (16000g 4˚C), the RNA pellet was washed twice with 70% ethanol, air dried for 5 min at room temperature and resuspended in 20μl water. To remove genomic DNA contamination, 3μl RNA was treated with 2 units of RNase-free DNase I (NEB; Massachusetts, USA) at 37˚C for 15 min. RNA was then extracted with phenol-chloroform and precipitated as described above.

## Quantitative PCR (qPCR)

1μl RNA from each fraction was reverse transcribed using an Omniscript RT kit (Qiagen, Hilden, Germany) with random hexamers as primers according to the manufacturer's protocol. For normalisation, 2ng FLuc RNA (Promega, Wisconsin, USA) was added to the reverse transcription reaction. 1μl of cDNA (diluted 1:10) was used for qPCR. Four trypanosomal mRNAs (COX IV, COX VIII, Elongation factor 1a, alpha-tubulin) and FLuc were quantified; the primers used are listed in S4 Table. qPCR was performed using MESA Green qPCR MasterMix Plus for SYBR assay (Eurogentec, Liege, Belgium) in the ABI Prism 7000 sequence detection system (Applied Biosystems, California, USA). The abundance of a target mRNA in each fraction was calculated as described [84].

## Nascent protein synthesis

Thirty millilitres of parasite cultures grown to a density of 5 x $10^6$ cells/ml were pulsed for 4h with 50μM L-homopropargylglycine (Jena Biosciences, Jena, Germany). Cells were pelleted and washed once with PBS. The cell pellet was then resuspended in lysis buffer (50mM Tris, pH 8.0 containing 1% SDS, 0.5% NP-40 and EDTA-free protease inhibitor cocktail (Roche, Basel, Switzerland)) and sonicated 3 times for 10s with a Branson Digital Sonifier at 10% amplitude, interspersed with 30-second intervals on ice. The lysate was clarified by centrifugation at 16,000g in an Eppendorf microcentrifuge at 4˚C for 5 min. Proteins were precipitated with chloroform-methanol [85] and solubilised in 50mM Tris, pH 8.0 containing 1% SDS. Protein concentrations were measured with a Pierce BCA protein assay Kit (ThermoFischer Scientific, Waltham, MA, USA) and 100μg of protein was used to add azide-biotin by copper-

dependent click-it chemistry. The click-it reaction was performed in 100mM sodium phosphate buffer pH 7.0, containing 5mM $CuSO_4$, 5mM Tris-hydroxypropyltriazolylmethylamine, 100μM azide-biotin and 100mM ascorbic acid for 30 min at room temperature. Proteins were then precipitated with chloroform-methanol to remove free azide-biotin molecules and resuspended in 100μl 50mM Tris pH 8.0 containing 1% SDS. To this, 900μl of IP buffer was added and biotinylated proteins were purified using streptavidin conjugated Dynabeads (Dynabeads M280, ThermoFischer Scientific, Waltham, MA, USA) and subjected to immunoblotting.

## Quantitative proteomics and RNA-Seq

**Quantitative proteomics.**   Approximately $1.5 \times 10^8$ cells were harvested by centrifugation (1400g, 8min), washed two times with PBS and pellets were flash frozen in liquid nitrogen. Frozen cell pellets were thawed and lysed in 100μl 8 M urea, 100mM Tris/HCl pH 8 by incubation at room temperature for 15 min with cycles of strong agitation every five minutes. In order to break down DNA, 33.3 μL of 100 mM Tris/HCl, 10 mM magnesium di-chloride pH 8.0 and 3.3 units of Universal Nuclease (Pierce) were added and incubated for 30 min at 37˚C. Proteins were reduced (10mM DTT, at 37˚C for 30 min) and alkylated (50mM iodoacetamide at 37˚C for 30 min in the dark), and precipitated with 5 volumes of cold acetone overnight at -20˚C. According to the protein concentration as measured by Qubit protein assay (Thermo-Fisher Scientific) prior to precipitation, the dry protein pellet was reconstituted in 8M urea, 50mM Tris/HCl pH 8 to a concentration of 2 mg/ml. An aliquot corresponding to 10μg protein was diluted with 20mM Tris/HCl, 2mM calcium di-chloride pH 8.0 to a final urea concentration of 1.6 M prior to digestion with trypsin at a protein-to-trypsin ratio of 50:1 (w/w) for 6 hours at 37˚C. The digests were acidified with trifluoroacetic acid and analysed by liquid chromatography (LC)-MS/MS (EASY-nLC 1000 coupled to a QExactive HF mass spectrometer, ThermoFisher Scientific) with 2 replicate injections of 5μl digest each. Peptides were trapped on a μPrecolumn C18 PepMap100 (3μm, 100 Å, 300μm×5mm, ThermoFisher Scientific, Reinach, Switzerland) and separated by backflush on a C18 column (Magic AQ, 5μm, 100 Å, 75μm×30 cm) by applying a 120-minute gradient of 5% acetonitrile to 40% in water, 0.1% formic acid, at a flow rate of 350 nl/min. The Full Scan method was set with resolution at 60,000 with an automatic gain control (AGC) target of $10^6$ and maximum ion injection time of 50 ms. The data-dependent method for precursor ion fragmentation was applied with the following settings: resolution 15,000, AGC of 1E05, maximum ion time of 110 milliseconds, mass window 1.6 *m/z*, collision energy 27, under fill ratio 1%, charge exclusion of unassigned and 1 + ions, peptide match preferred, and dynamic exclusion of fragmented precursors for 20 s, respectively.

Mass spectrometry data was interpreted with MaxQuant software (version 1.5.4.1) against the forward and reversed *T. brucei* TREU927 protein sequence fasta file (release version 39) using the following settings: strict trypsin cleavage rule allowing up to three missed cleavage sites, variable modifications of acetylated protein N-termini and oxidation of methionine and fixed carbamidomethylation of cysteines; peptide and fragment mass tolerances of 10 and 20 ppm; match between runs of the same cell line activated, but not between different cell types, in order to avoid over-interpretation. Identified peptides and proteins were filtered to a 1% false discovery rate based on reversed database matches and a minimum of two razor or unique peptides were set as acceptance criteria for protein group identification. Statistical testing of differentially expressed proteins was performed in R using a top3 intensity approach as a surrogate for protein abundance. Briefly, iTop3 values were calculated from the sum of the intensity of the 3 most intense peptides of each Leading Razor Protein. Before performing the sum, the peptide intensities were median-normalised to the global median, followed by

imputation. Imputation values were drawn from a gaussian distribution of width 0.3 centered at the sample distribution mean minus 1.8x sample standard deviation. In order to perform statistical tests, missing iTop3 values were further imputed at the protein level, using the same logic as for peptide imputation but with a down-shift of 2.5 standard deviations.

RNA-seq approximately 2 x $10^7$ cells were harvested at each time point and total RNA was extracted with acid guanidinium thiocyanate as described [86]. Genomic DNA contamination was eliminated by DNase treatment, as described above, and RNA was quantified using a Qubit RNA HS assay kit (Invitrogen, Eugene, Oregon, USA). cDNA libraries were prepared from poly(A)-selected RNA and sequenced at Fasteris (Geneva, Switzerland). Reads of 150bp at a sequencing depth of 20 million reads per sample were generated. Reads were mapped to the *T. b. brucei* (TREU927) reference genome version 5 and analysed as described previously [29], except that the read counts were extracted using the "featureCounts" tool available on the Galaxy platform [87]. Raw read files are deposited at the European Nucleotide Archives (ENA) under study PRJEB38690.

## Supporting information

**S1 Fig. Relationship of Alba proteins (A)** ClustalW protein sequence alignment of Alba3 and Alba4. Dark blue regions indicate sequence identity. The yellow bar indicates the RGG/ RG repeats present in the C-terminal regions. **(B)** Percentage protein identity between pairs of *T. brucei* Albas determined by using Clustal Omega pairwise alignment matrix [88,89]. (TIF)

**S2 Fig. Localisation of Alba proteins in different life cycle stages of *T. brucei* Localisation of Alba1-4 in slender forms, stumpy forms and early and late procyclic forms. Immunofluorescence analyses were performed using *in situ* N- terminal HA-tagged Alba cell lines.** Cells were incubated with anti-HA. Scale bar, 5μm. (TIF)

**S3 Fig. Effect of knockdown of Albas on growth.** RNAi was performed in slender bloodstream forms (BSF), early procyclic forms (early PCF) and late procyclic forms (late PCF). Cumulative growth of cells was monitored for 6 days without RNAi induction (-Tet) or with RNAi induction (+Tet) by tetracycline. Error bars, mean ±SD (n = 3). Efficiency of knockdown by RNAi was assessed by Western blot analysis on day 4 after RNAi induction. RHS1 served as a loading control. (TIF)

**S4 Fig. Confirmation of knockouts. (A)** Schematic representation of Alba 3+4 loci. **(B)** & **(D)** PCR confirmation of knockouts in Alba3KO, Alba4KO and Alba3&4KO+inducible Alba3 expression (A3&4KO+iA3) cell lines. Genomic DNA was isolated from individual cell lines and PCR was performed using primers indicated in the diagram in **(A). (C)** & **(E)** Western blot analyses of Alba3KO and Alba4KO. RHS1 served as a loading control. HOM, homology arms; CDS, coding sequence; Blast$^R$, blasticidin resistance gene; Puro$^R$, puromycin resistance gene. The sequences of the numbered primers (P1-11) are given in S4 Table. (TIF)

**S5 Fig. Analysis of Alba3 expression in differentiating cells.** Western blot analysis of Alba3 expression in AnTat 90–13, Alba3KO and Alba3cKO (-/+Tet) during differentiation. Alba3 expression was induced in Alba3cKO at different time points from the start of differentiation. RHS1 served as a loading control. (TIF)

**S6 Fig. GO term analysis.** GO term analysis of significantly down-regulated proteins (p value <0.01, fold change $\geq$ 2) in Alba3KO compared to AnTat 90–13 on days 0, 2 and 4 of differentiation. The x-axis represents the number of proteins and the y-axis shows GO categories. (TIF)

**S1 Table. Proteome of AnTat 90–13 and Alba3KO (Excel file).**
(XLSX)

**S2 Table. Primers and restriction sites (RE) used for construct preparation and RE site(s) used for linearisation of plasmids for transfection into trypanosomes.**
(TIFF)

**S3 Table. Antibody dilutions used in this study.**
(TIF)

**S4 Table. Primers used to confirm knockouts and qPCR primers.**
(TIFF)

**S1 Data. Excel spreadsheet containing, in separate sheets, the underlying numerical data for Figure panels: 1C, 1D, 2A, 2C, 2D, 2E, 2F, 3A, 3B, 3C, 5A, 5B, 6A, 6C, 6D, 7A, 7B, 8A, 8B, 9A, 9B, 9C, 9D and S3 and S6 Figs.**
(XLSX)

## Acknowledgments

We thank Ruth Etzensperger for her constructive comments on the manuscript, Keith Matthews for providing PAD1 and TbPIP39 antisera, André Schneider for providing cox IV antiserum and Jay Bangs for anti-VSG antiserum. Michal Domanski is thanked for help with polysome profiling experiments and Norbert Polacek for helpful discussions.

## Author Contributions

**Conceptualization:** Shubha Bevkal, Arunasalam Naguleswaran, Isabel Roditi.

**Formal analysis:** Shubha Bevkal, Manfred Heller, Isabel Roditi.

**Funding acquisition:** Isabel Roditi.

**Investigation:** Shubha Bevkal, Arunasalam Naguleswaran, Ruth Rehmann, Marcel Kaiser, Manfred Heller.

**Methodology:** Shubha Bevkal, Arunasalam Naguleswaran, Isabel Roditi.

**Project administration:** Isabel Roditi.

**Supervision:** Isabel Roditi.

**Validation:** Shubha Bevkal, Arunasalam Naguleswaran, Isabel Roditi.

**Visualization:** Shubha Bevkal.

**Writing – original draft:** Shubha Bevkal, Isabel Roditi.

**Writing – review & editing:** Shubha Bevkal, Arunasalam Naguleswaran, Isabel Roditi.

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
