## [Decision Letter · Decision Letter 0]

15 Sep 2020

Dear Prof. Roditi,

Thank you very much for submitting your manuscript "An Alba-domain protein required for proteome remodelling during trypanosome differentiation and host transition" for consideration at PLOS Pathogens. I apologise the manuscript has taken some time to review. As with all papers reviewed by the journal, your manuscript was reviewed by members of the editorial board and by several independent reviewers. In light of the reviews (below this email), we would like to invite the resubmission of a significantly-revised version that takes into account the reviewers' comments.

All of the reviewers appreciated the value of the findings in the manuscript, identifying a protein important at a well defined stage in the differentiation of trypanosomes as they develop from bloodstream to procyclic forms. However, they each also ask for revisions to support the overall findings of the work. In particular I draw your attention to the reviewers' comments that ask for clarification, additional evidence or rewording with respect to:

a. the phenotype of Alba3 knockout cells during the time-course of differentiation to procyclic forms: where do they arrest- in the first cell cycle, or thereafter? Is VSG lost with normal kinetics? This will help position the point at which Alba 3 is important in differentiation.

b. How fly infectivity and parasite progression develops over time in the tsetse fly and how this relates to the kinetics of the differentiation failure in vitro?

c. Confirmation that the protein synthesis assay used is consistent with more conventional assays using 35S methionine

d. Consideration of the ribosomal profile and biosynthesis in different life cycle stages and in the Alba 3 KO

e. Statistical analysis of the results to support your findings and to allow replacement of phrases that do not have scientific meaning with those that do.

We cannot make any decision about publication until we have seen the revised manuscript and your response to the reviewers' comments. Your revised manuscript is also likely to be sent to reviewers for further evaluation.

Sincerely,

Keith R. Matthews

Guest Editor

PLOS Pathogens

David Horn

Section Editor

PLOS Pathogens

Kasturi Haldar

Editor-in-Chief

PLOS Pathogens

orcid.org/0000-0001-5065-158X

Michael Malim

Editor-in-Chief

PLOS Pathogens

orcid.org/0000-0002-7699-2064

Reviewer's Responses to Questions

**Part I - Summary**

Reviewer #1: The T. brucei genome encodes 4 different Alba domain containing proteins, Alba1 to 4. Two of these, Alba 3 and 4, also contain a nucleic acid binding motif. The four Albas have previously been shown to localise to the cytoplasm of procyclic forms. The experiments in this manuscript start with an investigation of the subcellular localisation in bloodstream forms and make the discovery that Alba3 and 4 are concentrated in the Stumpy Regulatory Nexus in stumpy bloodstream forms. This finding suggests that they may play a role in the differentiation of mammalian bloodstream forms to insect procyclic forms and this was tested using first RNAi and then gene deletion. Knockdown of Alba3 produced a growth defect in procyclic forms but not in bloodstream forms, the other Albas did not and the remaining experiments concentrate on Alba3.

As bloodstream Alba3 -/- cells were fine in culture and in a murine model infection but stopped proliferating in differentiation to procyclic forms was induced in culture. This growth arrest was rescued by an inducible add back. These experiments indicate that Alba3 is involved for the differentiation to mature procyclic forms. Experimental infections of tsetse flies indicated that Alba3 was not necessary for differentiation and tsetse infection but Alba3-/- cells infected fewer flies and produced weaker infections.

Detailed analysis of the kinetics of differentiation showed that Alba3-/- cells were able to differentiate but did so more slowly than the parental. After differentiation, the procyclics forms stopped proliferating unless Alba3 was provided within 4 days of initiating differentiation.

Subsequent experiments investigated the molecular differences between parental and Alba3-/- cells. Mass spec comparison of the proteomes of parental and Alba3-/- indicated a reduction/absence of a large group of mitochondrial proteins in Alba3-/- cells. These differences were not reflected in mRNA levels suggesting a defect in translation.

Overall, the experiments in the paper provide strong evidence that Alba3 is necessary for the correct programme of translation on differentiation of stumpy bloodstream forms to procyclic forms and to establish a stable mature procyclic phenotype.

It is one of the first examples of a regulator of translation of a subset of mRNAs in tryps and will be of interest to a wide audience.

Reviewer #2: This manuscript details how two predicted RNA binding proteins in Trypanosoma brucei contribute to life cycle development. The Alba proteins in T. brucei have been previously discovered and characterised to some extent, but this paper carries out a thorough analysis of the importance of, particularly, Alba 3. This involves carefully executed experiments whereby Alba 3 expression is inducibly knocked down by RNAi, or full gene deletion achieved - with conditional re-expression. The resulting phenotypes and mRNA and proteome changes are characterised in the resulting lines, as are the detailed consequences for different stages of development. The quality of the experimental work and clarity of writing is characteristic of this group and the paper nicely shows that Alba 3 influences the efficiency of transition from early differentiated procyclic forms (i.e. soon after their generation from stumpy forms) to proliferative established early procyclic forms. The evidence supports their contention that Alba 3 affects proteome remodelling early in differentiation and that Alba 3 and Alba 4 show some functional redundancy in the bloodstream but not procyclic forms. Although direct versus indirect effects of Alba 3 on translation in early differentiation have not been pinned down, this is very difficult to establish and does not diminish the overall precision of the analysis reported. I have only a few requests for clarification and suggestions for improvement.

Reviewer #3: In this manuscript the authors characterize the family of Alba-domain proteins (Alba1-4) from Trypanosoma brucei. RNAi of each individual Alba genes show that they are not essential for growth. However, Alba3 appears to be important during the differentiation of stumpy cells into procyclics. This in vitro observation was then validated in vivo by infecting tsetse flies, which show a significant reduction in both the midgut and the proventriculus infection rates. The authors then show evidence that Alba3 is important to scape from translational repression, which they suggest impacts the formation of procyclic-specific mitochondrial respiratory complex proteins and the repression of some bloodstream-specific proteins.

The paper is well written and the strength of it is on the discovery that Alba3 is important for procyclic differentiation/colonization of the fly. However, two things concerns me: 1) Some of the findings and conclusions of this paper are similar to the ones reported by the same group in 2011 (Mani J., et al. 2011 PLoS One), so I wonder about how much progress has been made and, 2) the finding that Alba3 is important for the transition between stumpies and PCF is exciting, but I feel that the Alba3 KO mutants should have been better characterized both in vitro and in vivo. For instance, why do the cells stop growing in vitro (defects in cell cycle?)? Do they have problems with motility, flagellum structure or organelle repositioning after differentiation? Likewise in the fly, the infections look lower in the Alba3 KO, but do they catch up with WT levels if scored later, for instance, at 21 dpi? Furthermore, any evidence of a SoMo phenotype given that PV infections are significantly lower? Last, but not least, why does the infectivity of the Alba3 AB cells are not included in this experiment?

**Part II – Major Issues: Key Experiments Required for Acceptance**

Reviewer #1: 1. Is there an inconsistency between the tsetse infection which in some flies goes on for >4 days and the in vitro differentiation which stops proliferation?

2. Figure 6A

Does Figure 6A infer that stumpy forms contain very few ribosomes as well as very few polysomes and the subunits and monosomes peaks are very small? In the methods equal numbers are cell equivalents are used yet the y-axis for stumps has a different scale. These observations should be compared with previous findings (for example doi: 10.1371/journal.pone.0067069)

The y-axes should be on the same scale

It is a major finding if the decrease in protein synthesis is procyclics is due to a loss of ribosomes. Probably unlikely but if correct (as opposed to a technical cause for the observation), could Alba3 be delaying ribosome biosynthesis?

Could the number of repeats be added to the figure legends, for example the polysome analysis

3. Figure 6C

Has a comparison of the patterns of protein synthesis of L-HPG and 35S-methionine been reported? It is a minor point but I would like to be convinced they are the same.

Reviewer #2: None

Reviewer #3: The co-localization of Alba3/4 with PIP39 in the STuRN is interesting, but this story kind of disappears from the paper and its significance is never discussed.

Does the delay in MSP expression in Alba3 KO cells suggests that removal of the VSG coat is also delayed in differentiating cells? Any evidence of that?

The fly infectivity of just WT cells is rather low (~40%) and so the comparison with Alba3 KO cells is difficult because of the overall low infection rate. In addition, if flies were scored between 12-15 dpi it means that some variation in the infection phenotype can be expected considering the negligible growth in vitro of the KO cells. This also relates with my previous comment on the infection phenotype at 21 dpi and whether the infection level may eventually catch up over time.

**Part III – Minor Issues: Editorial and Data Presentation Modifications**

Reviewer #1: The starting cell line carries a couple of transgenes and should be referred to as ‘parental’ not ‘wild type’

The authors use descriptors such as ‘critical’ and ‘pivotal’. These does not mean anything scientifically. They may consider using ‘necessary’ and ‘sufficient’ etc.

Reviewer #2: Line 51. Abstract (and elsewhere in the text): “We show that Alba 3 is key to escape from translational repression…”. This feels a little too strong given that translation resumes in Alba KO lines albeit with delayed kinetics. Perhaps tone down to something like “We show that Alba 3 is key to rapid/efficient escape from translational repression…”.

Line 100. The review reference cited is quite old- there are more recent examples (e.g. PMID: 31164043 from the same author).

Line 157 (and other places where quite subtle phenotypes are observed). Alba 3 KO creates slower growth of the cells- is this at the level of statistical significance? The effect seems clear but is subtle and so would benefit from statistical analysis. This is also true of several other assays in the manuscript where ‘representative’ results are shown from several biological replicates. There is inherent variability in developmental studies such as those reported and so it is OK to provide representative datasets, but there should be ways to normalise variability between datasets to provide a statistical/numerical assessment for treatment versus control groups.

Line 160. It is intriguing that Alba 4 escapes RNAi in bloodstream forms when co-depleted with Alba 3. Could the authors speculate on how this might occur? The RNA machinery still clearly functions because Alba 3 is depleted and the failure to knockdown is always Alba 4, rather than either Alba 3 or 4 in different cells in the population.

Line 171 (Figure 2A-D) In the in vitro developmental studies it is reported and clear that stumpy are still generated in the Alba3 KO line – so, indeed, Alba 3 is not needed for stumpy formation. But it would be helpful to quantitate that since delayed or inefficient onward differentiation to procyclic forms could be generated if the level of stumpy forms was quantitatively less than in wild type cells. The data in Figure 3 suggest that stumpy formation is quite efficient, so it is not a major concern. Nonetheless, since other subtle phenotypes are reported in the paper, this easily quantifiable developmental step could be detailed.

Line 182. Was the first cell cycle of the differentiating cells assessed? In other words, although kinetoplast repositioning clearly happens (albeit a bit delayed; Figure 3B), did the cells then progress to divide their kinetoplast and undergo mitosis? This would help to determine if the cells block after the first differentiation cell cycle is completed, or during the first cell cycle.

Line 192-199. The infectivity of the Alba 3 KOs in tsetse flies is clearly less than wild type cells on average (Figure 3F). But nonetheless some flies establish medium and heavy midgut infections with the KO parasites. How do the authors explain this variability in the relative infectivity of the KO cells? Also please specify how many biological replicates were carried out- otherwise low infectivity to flies by the Alba3 KO could be a result of the particular bloodstock used (e.g. it contained less stumpies or was otherwise less infective for reasons unrelated to Alba 3).

Line 255. The difference between WT and Alba3 KO cells on day 2 and day 4 are reported not to overlap with the slender proteome described in Reference 24. It would have been helpful to have a parallel slender proteome dataset to the WT vs KO analyses derived from the same laboratory because interlab variation in proteomic studies will inevitably show differences. Are there any such slender datasets available as part of the same WT/KO experiment to strengthen the overall statement/argument?

Line 304-306 (Figure 6C) Why is the protein synthesis in the induced Alba3 re-expression so much stronger than in wild type cells? Also, on Figure 6C it is reported tubulin levels are equivalent on day 2 and day 4 between the samples, but it seems a weaker in the day 2 samples with Alba3 KO without conditional rescue.

Reviewer #3: None

PLOS authors have the option to publish the peer review history of their article (what does this mean?). If published, this will include your full peer review and any attached files.

Reviewer #1: No

Reviewer #2: No

Reviewer #3: No
---

## [Editor Report · Decision Letter 1]

10 Dec 2020

Dear Prof. Roditi,

We are pleased to inform you that your manuscript 'An Alba-domain protein required for proteome remodelling during trypanosome differentiation and host transition' has been provisionally accepted for publication in PLOS Pathogens.

Best regards,

Keith R. Matthews

Guest Editor

PLOS Pathogens

David Horn

Section Editor

PLOS Pathogens

Kasturi Haldar

Editor-in-Chief

PLOS Pathogens

orcid.org/0000-0001-5065-158X

Michael Malim

Editor-in-Chief

PLOS Pathogens

orcid.org/0000-0002-7699-2064

I have been through the amendments to the paper in response to the referees' comments on the initial submission. All the issues raised have been adequately addressed.
---

## [Editor Report · Acceptance letter]

19 Jan 2021

Dear Prof. Roditi,

We are delighted to inform you that your manuscript, "An Alba-domain protein required for proteome remodelling during trypanosome differentiation and host transition," has been formally accepted for publication in PLOS Pathogens.

Best regards,

Kasturi Haldar

Editor-in-Chief

PLOS Pathogens

orcid.org/0000-0001-5065-158X

Michael Malim

Editor-in-Chief

PLOS Pathogens

orcid.org/0000-0002-7699-2064